# The origin of a primordial genome through spontaneous symmetry breaking

Nobuto Takeuchi[1,2], Paulien Hogeweg[2] & Kunihiko Kaneko[1]

The heredity of a cell is provided by a small number of non-catalytic templates—the genome. How did genomes originate? Here, we demonstrate the possibility that genome-like molecules arise from symmetry breaking between complementary strands of self-replicating molecules. Our model assumes a population of protocells, each containing a population of self-replicating catalytic molecules. The protocells evolve towards maximising the catalytic activities of the molecules to increase their growth rates. Conversely, the molecules evolve towards minimising their catalytic activities to increase their intracellular relative fitness. These conflicting tendencies induce the symmetry breaking, whereby one strand of the molecules remains catalytic and increases its copy number (enzyme-like molecules), whereas the other becomes non-catalytic and decreases its copy number (genome-like molecules). This asymmetry increases the equilibrium cellular fitness by decreasing mutation pressure and increasing intracellular genetic drift. These results implicate conflicting multilevel evolution as a key cause of the origin of genetic complexity.

[1] Department of Basic Science, Graduate School of Arts and Sciences, University of Tokyo, Komaba 3-8-1, Meguro-ku, Tokyo 153-8902, Japan. [2] Theoretical Biology and Bioinformatics Group, Utrecht University, Padualaan 8, 3584CH Utrecht, The Netherlands. Correspondence and requests for materials should be addressed to N.T. (email: takeuchi@complex.c.u-tokyo.ac.jp)

Heredity is an essential prerequisite for evolution and also is itself subject to evolution. Thus, understanding the evolution of heredity is a fundamental issue in evolutionary biology. The heredity of a modern cell is provided primarily by its genome. The genome of a modern cell has many notable features, but two stand out as being universal, with which the present study is concerned. First, a genome consists of molecules that serve as templates, some (but not all) of which provide information for producing catalysts, but they themselves do not serve as catalysts; therefore, there is a functional differentiation between templates and catalysts. Second, the per-cell copy number of the templates is much smaller than that of the catalysts encoded by the templates; therefore, there is also a copy-number differentiation between templates and catalysts.

However, these two features of modern-type heredity are believed to have been absent at the earliest stages of evolution. The RNA world hypothesis posits that the heredity of the first, primitive cell (protocell, for short) was provided by a population of dual-functional molecules serving as both templates and catalysts. This hypothesis raises a question: how did 'non-catalytic, small-copy-number templates providing information for producing catalysts' evolve? Or, more briefly, how did such genome-like molecules evolve? (Hereafter, the word 'genome-like' is used in this sense.)

A key to this question has been suggested by Szathmáry and Maynard Smith[1] (see also refs [2, 3]). According to their suggestion, the first, primordial form of genomes arose from functional asymmetry between complementary strands of replicating molecules, whereby one strand served as both a template and a catalyst, whereas the other served only as a template. Because it pays to produce more catalysts than templates, evolution would increase the copy number of the catalytic strand[3], commensurately decreasing that of the non-catalytic strand, hence the evolution of non-catalytic, small-copy-number templates, i.e., genome-like molecules.

The suggestion offered by Szathmáry and Maynard Smith raises two further questions. First of all, is the resemblance between such genome-like molecules and the genome as we know it more than purely formal? In other words, do such genome-like molecules have any special quality related to heredity, besides having the two features of a genome mentioned above? Second, what evolutionary mechanism can account for the postulated functional asymmetry? This mechanism has to allow for the fact that both complementary strands could in principle have catalytic activity[4] (see also Discussion). Having catalytic activity in both strands would be far more efficient (particularly in the absence of transcriptional or translational amplification). Thus, such sequences could be strongly favoured by natural selection, even if they might be rare within sequence space. Such selection could preclude the postulated asymmetry.

Here, we address both of these questions by considering a conflict between evolution at the cellular level and evolution at the molecular level. At the cellular level, evolution tends towards maximising the catalytic activities of molecules contained within protocells in order to maximise the growth rates of protocells (i.e., the evolution of intracellular catalytic cooperation). At the molecular level, however, an opposite trend emerges owing to a fundamental trade-off between templates and catalysts[5, 6]: evolution tends towards minimising the catalytic activities of the molecules in order to maximise their relative chance of replication within each protocell (i.e., the evolution of selfish replicators[7, 8]). Using individual-based modelling, we demonstrate the possibility that these conflicting tendencies of evolution operating at multiple levels induce spontaneous symmetry breaking between the complementary strands of the molecules, whereby one strand maintains its catalytic activity and increases its copy number (enzyme-like molecules), whereas the other strand completely loses its catalytic activity and decreases its copy number (genome-like molecules). Thanks to their small copy-numbers, these genome-like molecules experience increased intracellular genetic drift, which neutralises their evolutionary tendency to minimise the catalytic activities of their complements. Thereby, the genome-like

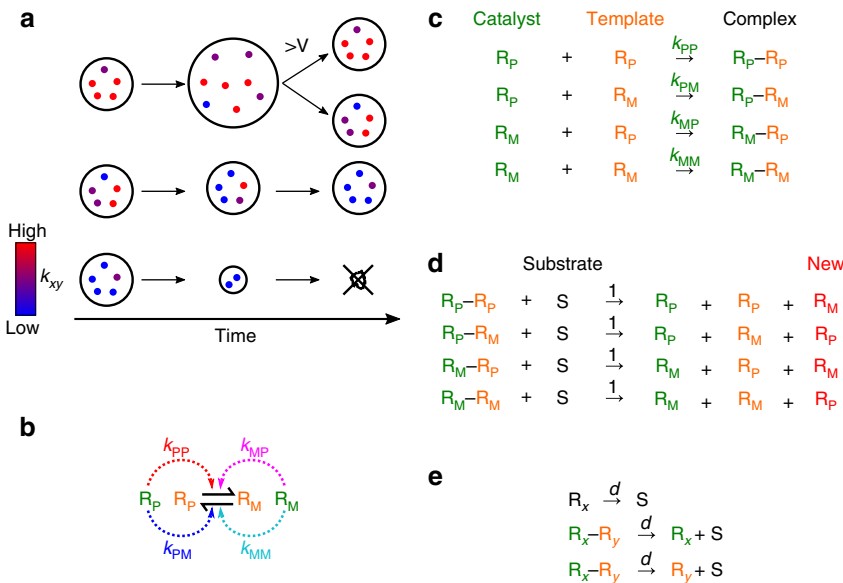

**Fig. 1** Schematic of the model. **a**. Protocells containing replicators (substrates not shown). Colours indicate the catalytic activity of replicators $k_{xy}$. A protocell with high-$k_{xy}$ replicators grows and divides (*top*); that with low-$k_{xy}$ replicators shrinks and dies (*bottom*). Replicators within a protocell evolve towards minimising $k_{xy}$ (*middle*). **b**. Schematic of replication and catalysis (see also **c** and **d**). *Solid arrows* indicate replication (template→product); *dotted arrows* indicate catalysis (catalyst⋯→reaction). **c**. Complex formation. $R_x$ denotes a replicator ($x \in \{P, M\}$); $R_x - R_y$, complex between $R_x$ serving as a catalyst (*green*) and $R_y$ serving as a template (*orange*) ($y \in \{P, M\}$). The complex formation rate is given by the $k_{xy}$ value of the catalyst. **d**. Replication. S denotes a substrate. A newly-synthesised replicator (*red*) is complementary to the template (*orange*), with its $k_{xy}$ values copied from the template with possible mutations. **e**. Decay. $d = 0.02$, unless otherwise stated

**Table 1 Notation**

| | Description | Default |
|---|---|---|
| $V$ | Protocell divides when it has $\geq V$ particles | $10^2 \sim 10^4$ |
| $m$ | Probability of mutation of $k_{xy}$ per replication | 0.01 |
| $N$ | Total number of particles in the system | $50V$ |
| $N_P$ | Total copy number of the plus strand | Variable |
| $N_M$ | Total copy number of the minus strand | Variable |
| $k_{xy}$ | Complex formation rate (see main text) | Evolvable |
| $k_P$ | $=k_{PP}=k_{PM}$ (when kinetic symmetry is imposed) | Evolvable |
| $k_M$ | $=k_{MP}=k_{MM}$ (when kinetic symmetry is imposed) | Evolvable |
| $d$ | Decay rate of replicators | 0.02 |
| $\delta$ | 'Mutation step size of $k_{xy}$' $\in (-\delta, \delta)$ | 0.05 |
| $\kappa$ | $\equiv (k_{PM}-k_{PP})/(k_{PM}+k_{PP})$ (kinetic asymmetry) | Evolvable |

molecules provide long-term stability to the genetic information of protocells.

## Results

**Model**. The model is described below in general terms (see Methods section for details). The model consists of two types of particles, replicators and substrates, which are partitioned into protocells (Fig. 1a). Replicators consume substrates to replicate (Fig. 1d). Substrates are generated via the decay of replicators (Fig. 1e). Thus, the total number of particles (i.e., replicators and substrates) $N$ is kept constant throughout a simulation (Table 1). Substrates freely diffuse across protocells, whereas replicators do not (both diffuse extremely rapidly within a protocell).

Replication is assumed to occur in two steps. First, two replicators form a complex, where one serves as a catalyst and the other as a template (Fig. 1c). Subsequently, the complex converts a substrate into a complementary copy of the template, and then the complex dissociates (Fig. 1d). This two-step replication incorporates the fact that replication takes a finite amount of time. During this time, a replicator serving as a catalyst is assumed to be incapable of simultaneously serving as a template. This assumption is based on the constraint, which is likely to exist in RNA molecules, that providing catalysis and serving as a template impose structurally incompatible requirements[6, 9]. Under the above assumption, a trade-off inevitably emerges: spending more time serving as a catalyst comes at the cost of spending less time serving as a template, with the consequence of inhibited self-replication[5]. Thus, serving as a catalyst is an 'altruistic' act, which puts a replicator at a relative selective disadvantage within a protocell. This condition is crucial as its removal drastically changes the outcome of the model (as described in Results under 'Conflicting multilevel evolution causes symmetry breaking').

Each replicator is assumed to be capable of serving as both a catalyst and a template, albeit not simultaneously as mentioned above. Thus, a pair of replicators can form two distinct complexes depending on which serves as a catalyst or template. These choices depend on the rates of complex formation (denoted by $k_{xy}$) as defined below.

For simplicity, replicators within a protocell were assumed to be closely similar or nearly complementary to each other in their sequences. Under this assumption, a major part of sequence heterogeneity among replicators exists in the difference between complementary strands. Accordingly, the model assumes two classes of replicators: the plus strand and minus strand (denoted by P and M, respectively). The two strands can potentially differ, both in their catalytic activity and in their affinities towards catalysts as templates. To encompass these possibilities, each replicator is assigned four rates of complex formation denoted by $k_{xy}$, where $x \in \{P, M\}$ and $y \in \{P, M\}$ (Fig. 1b). Among the four

$k_{xy}$ values of a given replicator, two values denote the rates at which this replicator, serving as a catalyst, forms a complex with any replicator serving as a template in the same protocell, where $x$ is the strand type of this replicator, and $y$ is that of the replicator serving as a template; the other two values apply to the complement of this replicator (Fig. 1c). Thus, with higher values of $k_{xy}$, there is a greater probability that a replicator or its complement serves as a catalyst.

When a new replicator is produced (Fig. 1d), its $k_{xy}$ values are copied from the template. Mutation occurs with probability $m$ per replication and changes the $k_{xy}$ values of the new replicator by adding a random number between $-\delta$ and $\delta$ ($m = 0.01$ and $\delta = 0.05$ unless otherwise stated). Each $k_{xy}$ value has an upper bound, but not a lower bound ($k_{xy} \leq 1$, unless otherwise stated); however, when $k_{xy} < 0$, the respective rate of complex formation is regarded as zero (for details, see Methods section under 'The mutation of $k_{xy}$ values').

As mentioned above, a replicator decreases its relative fitness within a protocell by serving as a catalyst. Therefore, replicators within each protocell tend to evolve towards minimising their $k_{xy}$ values—i.e., the evolution of selfish replicators. This tendency, however, is counteracted by evolution at the level of protocells as described below.

Protocells containing replicators with higher $k_{xy}$ values are at an advantage in competition for substrates because they consume substrates faster. Substrates are assumed to diffuse across protocells (extremely rapidly, for simplicity), whereas replicators do not diffuse at all. This difference in diffusion induces a net flow of substrates from protocells consuming substrates slowly (i.e., those containing low-$k_{xy}$ replicators) to those consuming them fast (containing high-$k_{xy}$ replicators)[10]. This flow causes protocells with high-$k_{xy}$ replicators to grow at the expense of those with low-$k_{xy}$ replicators. When the number of particles in a protocell exceeds $V$, the protocell is divided into two with its particles randomly distributed between the daughter cells (under this assumption, the number of protocells approximately tends to $2N/V$). As a consequence of differential growth and division, protocells tend to evolve towards maximising the $k_{xy}$ values of the replicators they contain. Therefore, evolution at the cellular level fosters catalytic cooperation between replicators within a protocell.

In summary, evolution operates at multiple levels with conflicting tendencies. Cellular-level evolution tends towards maximising $k_{xy}$ values, fostering catalytic cooperation between replicators; conversely, molecular-level evolution tends towards minimising $k_{xy}$ values, breeding catalytic exploitation within replicators.

**Evolution of genome-like molecules via symmetry breaking**. The model described above reveals that the complementary strands of replicators undergo spontaneous symmetry breaking for an intermediate range of cell sizes (as determined by $V$). The model was initialised with replicators whose complementary strands were functionally indistinguishable (specifically, $k_{PP} = k_{PM} = k_{MP} = k_{MM} = 1$ for every replicator; however, initial conditions are unimportant, as seen later in the section under 'The mechanism of functional symmetry breaking'). If $V$ is sufficiently small (Fig. 2a, $V \leq 650$), all $k_{xy}$ values are nearly maximised, so that the complementary strands remain indistinguishable. Conversely, if $V$ is sufficiently large (Fig. 2a, $V \geq 8000$), all $k_{xy}$ values are minimised, so that the system goes extinct. In this case, the complementary strands are again indistinguishable as they both become non-catalytic. Interestingly, if $V$ is in an intermediate range (Fig. 2a, $700 \leq V \leq 7500$), some $k_{xy}$ values are minimised, while the others assume distinct positive

values—i.e., the complementary strands undergo symmetry breaking (i.e., the average values of $k_{xy}$ are not invariant under the exchange of $P$ and $M$). This outcome was unexpected because there is no obvious selection pressure that can account for it: molecular- and cellular-level evolution either minimises or maximises all $k_{xy}$ values. The evolved asymmetry has two aspects: functional and kinetic. In the functional aspect, one strand (say P)

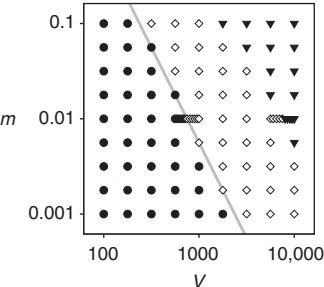

**Fig. 3** Phase diagram with respect to cell size ($V$) and mutation rate ($m$). The symbols indicate no symmetry breaking (*filled black circle*), symmetry breaking (*open diamond*), and extinction (*downward pointing triangle*). The boundary between ● and ◇ approximately follows the scaling relationship $mV \propto 1/V$ (*grey line*). The parameters were the same as in Fig. 2

remains catalytic, whereas the other (M) loses catalytic activity altogether, thus serving only as a template (i.e., $k_{Py} > 0$ and $k_{My} = 0$, where $y \in \{P, M\}$). In the kinetic aspect, the catalytic strand is produced more rapidly than the non-catalytic strand ($k_{PP} < k_{PM}$), resulting in a smaller copy number for the non-catalytic strand ($N_M/N_P \approx 3/7$ for $V > 1000$, where $N_x$ is the copy number of $x$). These two aspects of the symmetry breaking together produce non-catalytic, small-copy-number templates— i.e., genome-like molecules. Which of the two complementary strands evolve into genome-like molecules is entirely arbitrary; however, for the sake of presentation, genome-like molecules are hereafter taken to be always the minus strand (if they evolve). In summary, the model displays the three phases: (1) for small $V$, the complementary strands remain symmetric; (2) for intermediate $V$, the complementary strands undergo symmetry breaking and; (3) for large $V$, the system goes extinct.

To ascertain the robustness of the above results, we examined two additional methods of mutating $k_{xy}$ values (see Methods section under 'The mutation of $k_{xy}$ values'). In the first method, the reflecting boundary condition was imposed at $k_{xy} = 0$. In the second method, $k_{xy}$ was mutated in a logarithmic scale. In both cases, the existence of the three phases was confirmed, indicating the robustness of the results (Supplementary Fig. 1). Also, the results remain essentially the same when the model incorporates the continual emergence through mutation of parasitic replicators providing no catalysis (Supplementary Fig. 2).

**Conflicting multilevel evolution causes symmetry breaking**. The fact that the symmetry breaking occurs for an intermediate range of $V$ suggests that a key cause of the symmetry breaking is conflicting multilevel evolution. As described elsewhere[11], $V$ determines the relative rates of molecular-level evolution (i.e., the evolution of replicators within each protocell) and cellular-level evolution (i.e., the evolution of protocells). Decreasing

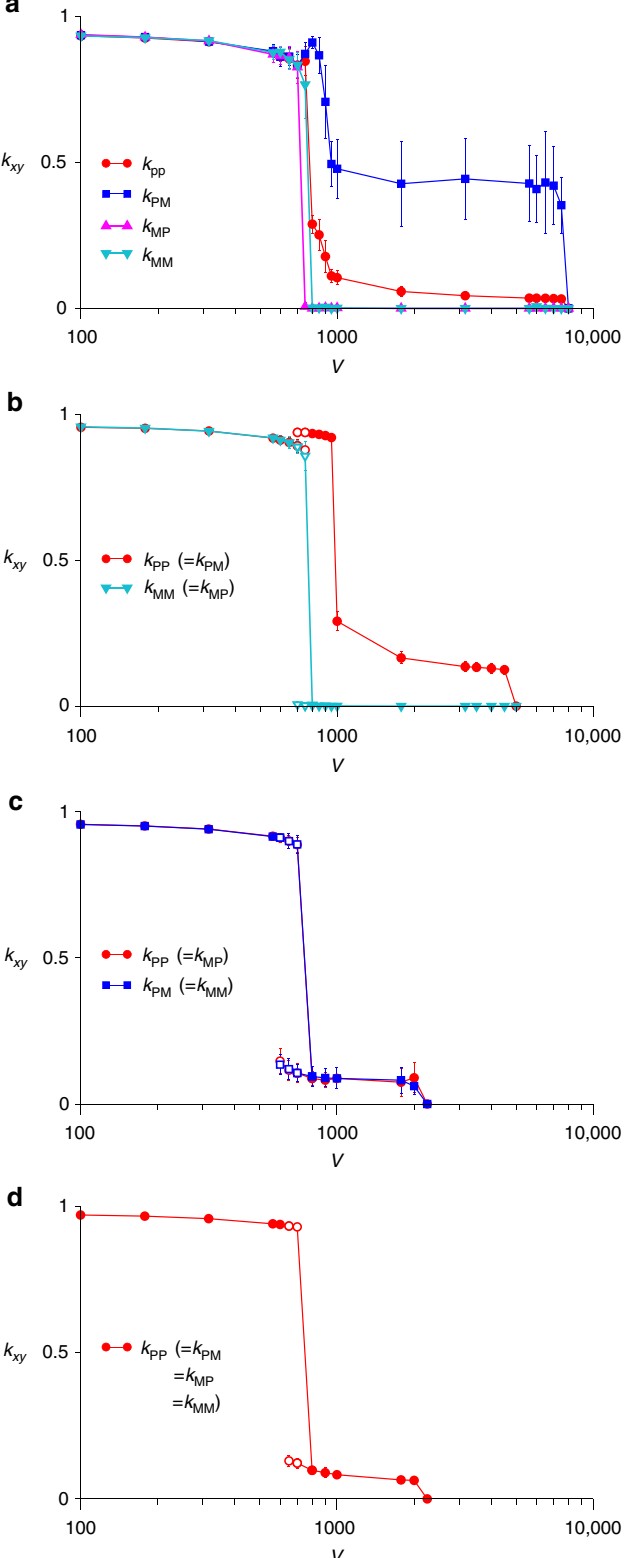

**Fig. 2** Equilibrium average catalytic activities ($k_{xy}$) as functions of cell size ($V$). The values of $k_{xy}$ were first averaged over all replicators at each time point. Then, the average (symbols) and s.d. (error bars) of $k_{xy}$ overtime were calculated after equilibration. The *open symbols* indicate metastable states (i.e., states reachable from different initial conditions with no state transition observed within $10^7$ time steps; a replicator decays approximately with a probability $d$ per time step). For visibility, the non-catalytic strand (if it evolves) was assumed to be always the minus strand. Parameters: $m = 0.01$, $d = 0.02$, $N = 50V$. **a**. The full model (no symmetry imposed). **b**. The model in which kinetic symmetry is imposed (i.e., $k_{PP} = k_{PM}$ and $k_{MP} = k_{MM}$). **c**. The model in which functional symmetry is imposed (i.e., $k_{PP} = k_{MP}$ and $k_{PM} = k_{MM}$). **d**. The model in which both kinetic and functional symmetry are imposed (i.e., $k_{PP} = k_{PM} = k_{MP} = k_{MM}$)

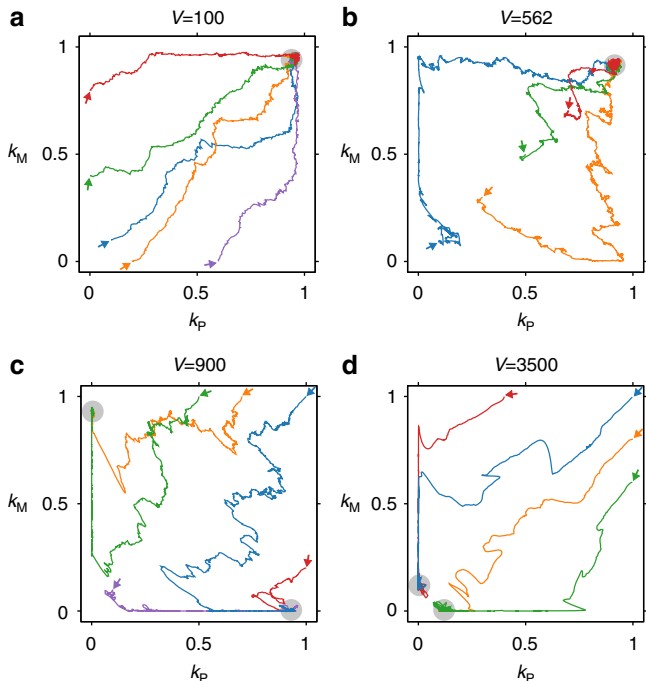

**Fig. 4** Evolutionary trajectories of the catalytic activities of the plus and minus strands ($k_P$ and $k_M$, respectively). The values of $k_P$ and $k_M$ were averaged over all replicators at each time point. The *arrows* indicate different initial conditions; *grey* circles indicate equilibria. The values of $V$ are shown above each graph: a. $V = 100$; b. $V = 562$; c. $V = 900$; d. $V = 3500$. The other parameters were the same as in Fig. 2. The data were obtained with the model in which kinetic symmetry is imposed (i.e., $k_{PP} = k_{PM}$ and $k_{MP} = k_{MM}$) so that evolutionary trajectories can be depicted in a plane

$V$ decelerates molecular-level evolution because it decreases the amount of mutational input per protocell. On the other hand, decreasing $V$ accelerates cellular-level evolution because it increases the amount of variation between protocells in terms of intracellular average $k_{xy}$ values (this occurs because decreasing $V$ increases intracellular genetic drift and the chance of unequal cell division). Therefore, if $V$ is sufficiently small, cellular-level evolution is expected to dominate over molecular-level evolution, maximising $k_{xy}$ values (Fig. 2a $V \leq 650$). Conversely, if $V$ is sufficiently large, molecular-level evolution is expected to dominate over cellular-level evolution, minimising $k_{xy}$ values (Fig. 2a $V \geq 8000$). For an intermediate range of $V$, evolution at neither level dominates over the other, a situation in which conflicting multilevel evolution ensues. The above consideration suggests that conflicting multilevel evolution is a key cause of the symmetry breaking.

The involvement of conflicting multilevel evolution implies that the symmetry breaking should occur only for an intermediate range of $m$ (mutation rate) because like $V$, $m$ affects the amount of mutational input per protocell. This expectation was confirmed by a phase diagram in terms of $m$ and $V$ (Fig. 3). This diagram additionally reveals an approximate scaling relationship

$$mV \propto 1/V$$

for the onset of the symmetry breaking (Fig. 3, grey line). This scaling relationship permits the following interpretation[11]. The left hand side ($mV$) is proportional to the amount of mutational input per protocell and, therefore, proportional to the rate of molecular-level evolution. The right hand side ($1/V$) is proportional to the amount of variation between protocells and,

therefore, proportional to the rate of cellular-level evolution. The above scaling relationship thus gives a further support to the hypothesis that conflicting multilevel evolution is a key cause of the symmetry breaking.

To confirm the above hypothesis, we examined the effect of removing conflicting multilevel evolution. Specifically, we modified the model by assuming that replication occurs in one step instead of two (i.e., assuming that replication is instantaneous). Under this model, serving as a catalyst no longer incurs an immediate fitness cost to a replicator[5]. Therefore, molecular-level evolution neither minimises nor maximises $k_{xy}$ values (i.e., neutral), whereas cellular-level evolution still tends towards maximising $k_{xy}$ values; i.e., evolution at the two levels are not in conflict with each other. This model displays no symmetry breaking for the examined range of $V$ ($V \leq 10^5$), confirming that conflicting multilevel evolution is a key cause of the symmetry breaking (Supplementary Fig. 3).

**Functional symmetry breaking induces kinetic symmetry breaking.** To investigate the mechanisms of the symmetry breaking, we sought to disentangle its functional and kinetic aspects. To this end, we imposed either functional or kinetic symmetry on the model. Specifically, kinetic symmetry was imposed by setting $k_{PP} = k_{PM}$ and $k_{MP} = k_{MM}$ for each replicator so that the plus strand and minus strand were replicated at identical rates. This constraint causes correlations between $k_{xy}$ values and, thereby, increases the amount of variation generated by mutation. However, this effect was cancelled out by modifying mutation step-size distributions (for details, see Methods section under 'The correction for mutation step-size distributions'). Even with this constraint, we found that replicators still undergo functional symmetry breaking if $V$ is sufficiently large (Fig. 2b, $V \geq 800$). This result indicates that the functional symmetry breaking is independent of kinetic symmetry breaking.

Next, functional symmetry was imposed on the model by setting $k_{PP} = k_{MP}$ and $k_{PM} = k_{MM}$ for each replicator so that the plus strand and minus strand had identical catalytic activities. With this constraint, we found that replicators no longer undergo symmetry breaking (Fig. 2c). This result indicates that kinetic symmetry breaking hinges on functional symmetry breaking.

Taken together, the above results indicate that functional symmetry breaking is logically prior to kinetic symmetry breaking. That the latter follows from the former is easy to understand. Because it pays to produce more catalysts than templates for protocells, functional symmetry breaking creates cellular-level selection pressure to increase the copy number of the catalytic strand, thereby inducing kinetic symmetry breaking[1, 3]. Together with this consideration, the above finding led us to focus on the mechanisms of functional symmetry breaking.

Additionally, Fig. 2 shows that imposing symmetry on the model substantially reduces the range of $V$ within which protocells survive. Specifically, imposing kinetic symmetry reduces this range from $V \leq 7500$ to $V \leq 4500$ (Fig. 2a, b). Imposing functional symmetry, and therefore also kinetic symmetry, reduces this range from $V \leq 7500$ to $V \leq 2000$ (Fig. 2a, c and d). The reduction in the survival range of $V$ implies that the rate of molecular-level evolution is increased relative to that of cellular-level evolution. Therefore, the above results suggest that both the functional and kinetic symmetry breaking stabilise protocells against the catalytic deterioration of replicators due to molecular-level evolution.

The last statement is corroborated by the fact that imposing functional or kinetic symmetry on the model decreases the equilibrium mean fitness of protocells for the ranges of $V$ within which protocells survive, where the fitness is measured by the

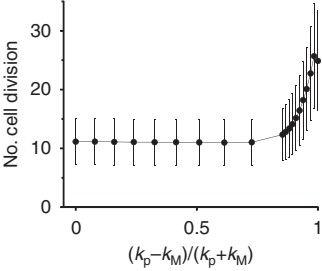

**Fig. 5** The average number of cell divisions a protocell undergoes along its line of descent as a function of the degree of functional asymmetry, $(k_P - k_M)/(k_P + k_M)$. Error bars indicate s.d. The initial fitness of protocells, measured by the expected fraction of replicators within a protocell, was kept constant for the different values of $(k_P - k_M)/(k_P + k_M)$ (for details, see Methods section under 'The measurement of the number of cell divisions'). Parameters: $m = 0.01$, $d = 0.0632$. $V = 1700$

fraction of particles being replicators (Supplementary Fig. 4). Note, however, that the equilibrium mean fitness of protocells is always a monotonically decreasing function of $V$, irrespective of whether or not any symmetry is imposed on the model (Supplementary Fig. 4).

**The mechanism of functional symmetry breaking**. To gain an insight into the mechanism of the functional symmetry breaking, we examined the evolutionary trajectories of $k_{xy}$ values. For simplicity, kinetic symmetry breaking was prevented as before by setting $k_{PP} = k_{PM}$ (denoted by $k_P$) and $k_{MP} = k_{MM}$ (denoted by $k_M$). The trajectories of $k_P$ and $k_M$ were computed using various initial conditions for different values of $V$ (Fig. 4).

When $V$ is much smaller than the critical value (Fig. 4a, $V = 100$), trajectories climb straight up until they hit phase-space boundaries, indicating that cellular-level evolution dominates over molecular-level evolution.

When $V$ is much larger than the critical value (Fig. 4d, $V = 3500$), trajectories fall straight down until they hit the boundaries, indicating that molecular-level evolution dominates over cellular-level evolution.

The above results are in agreement with the statement made in the previous section that increasing $V$ accelerates and decelerates molecular- and cellular-level evolution, respectively. More revealing insights are obtained when $V$ is close to the critical value:

When $V$ is slightly below the critical value (Fig. 4b, $V = 562$), some trajectories reach equilibrium via the boundary at $k_P = 0$ or $k_M = 0$, instead of climbing straight up.

When $V$ is slightly above the critical value (Fig. 4c, $V = 900$), trajectories first fall down towards the boundary at $k_P = 0$ or $k_M = 0$. After reaching the boundaries, the trajectories are reoriented so as to climb up along the boundaries.

To explain the above results, particularly the reorientation of the trajectories, we hypothesised that the rate of molecular-level evolution decreases near the phase-space boundary at $k_P = 0$ or $k_M = 0$, an effect that could reverse the power relationship between molecular- and cellular-level evolution. In other words, molecular-level evolution decelerates if the functional symmetry breaking occurs. This hypothesis is in line with the fact that the functional symmetry breaking stabilises protocells against molecular-level evolution (Fig. 2).

To test the above hypothesis, we measured the rate of molecular-level evolution as a function of the degree of functional asymmetry. The rate of evolution was gauged by the maximum number of cell divisions a protocell undergoes along its line of descent, as described below (for details, see Methods section

under 'The measurement of the number of cell divisions'). One protocell was continuously followed while it competed against a large, stationary population of protocells. When the protocell divided, one of its daughters was randomly chosen and continuously followed. As a protocell grows and divides along its line of descent, its internal replicators inevitably evolve towards minimising their $k_{xy}$ values (i.e., molecular-level evolution). Consequently, the protocell is put at a competitive disadvantage and eventually ceases to reproduce (Supplementary Fig. 5a). Therefore, the slower the evolution of internal replicators, the greater the number of cell divisions a protocell undergoes along its line of descent. The number of division was measured as a function of the degree of functional asymmetry defined as $(k_P - k_M)/(k_P + k_M)$. The results show that the number of division increases with $(k_P - k_M)/(k_P + k_M)$ (Fig. 5), indicating that the functional symmetry breaking decelerates molecular-level evolution.

The deceleration of molecular-level evolution near the phase-space boundaries explains why conflicting multilevel evolution induces functional symmetry breaking. For sufficiently large values of $V$, molecular-level evolution dominates over cellular-level evolution, so that $k_P$ and $k_M$ are decreased. The rates of this decrease, however, cannot be precisely identical because evolution is inherently stochastic. Suppose $k_M$ is decreased faster (in the following argument, P and M are exchangeable). As $k_M$ approaches zero before $k_P$ (i.e., as functional asymmetry arises), molecular-level evolution decelerates, shifting the balance of power in favour of cellular-level evolution. Consequently, cellular-level evolution increases $k_P$ to a level that is unattainable without the deceleration of molecular-level evolution. If, at this point, $k_M$ were increased, molecular-level evolution would accelerate, pushing the system back to the boundary. Therefore, the only states in which a stable balance can be achieved between molecular- and cellular-level evolution are on the boundary at $k_M = 0$. That is why the evolutionary trajectories stay on the boundary, hence functional symmetry breaking. In short, functional symmetry breaking occurs because it buffers a protocell against molecular-level evolution.

Why molecular-level evolution decelerates near the phase-space boundaries can be explained by a decrease in the amount of variation generated by mutation. According to Fisher's fundamental theorem, the rate of molecular-level evolution is proportional to the variance of the fitness of replicators within a protocell. Let the fitness of a replicator be denoted by $f(k_P, k_M)$. Because the mutations of $k_P$ and $k_M$ are small and uncorrelated, the amount of variation in $f(k_P, k_M)$ due to mutation can be approximated by

$$\sigma_f^2 \approx \left(\frac{\partial f}{\partial k_P}\right)^2 \sigma_{k_P}^2 + \left(\frac{\partial f}{\partial k_M}\right)^2 \sigma_{k_M}^2, \tag{1}$$

where the derivatives are evaluated at the per-cell average values of $k_P$ and $k_M$, and $\sigma_z^2$ is the variance of $z$ within a protocell. When the system approaches a phase-space boundary (say, $k_M \approx 0$), $\sigma_f^2$ decreases, because $\partial f/\partial k_M = 0$ if $k_M < 0$; hence, we see a deceleration in molecular-level evolution. The same argument holds when $k_{xy}$ is mutated by the alternative methods (see Methods section under 'The mutation of $k_{xy}$ values').

Based on the above argument, we searched for the minimal conditions required for functional symmetry breaking using a drastically simplified model. The results are described in the section under 'Sufficient conditions for functional symmetry breaking'. For the sake of continuity, however, we continue to describe the original model in the next section.

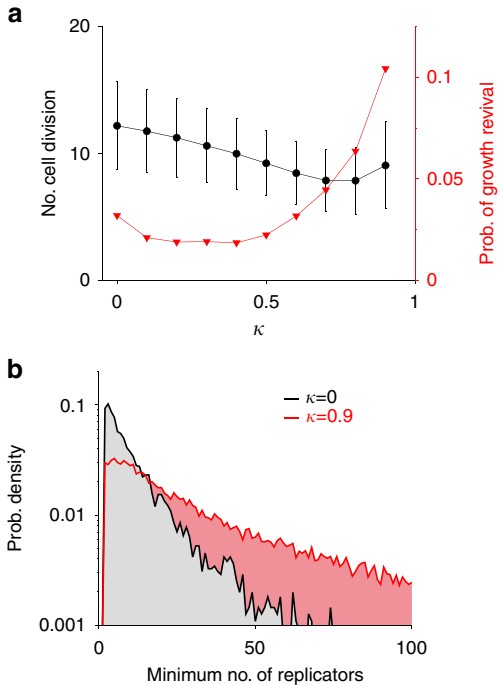

**Fig. 6** The benefit of kinetic symmetry breaking. **a**. The probability of bottleneck-induced growth restoration (*triangle*, right coordinate), and the average number of cell divisions a protocell undergoes along a single line of descent (*circles*, left coordinate; error bars, s.d.), as a function of the degree of kinetic asymmetry ($\kappa$). See Methods section for the details of measurements (under 'The measurement of the probability of growth restoration'). **b**. The empirical probability density of the minimum number of replicators in a protocell that successfully restored its growth

**The benefit of kinetic symmetry breaking**. We next asked why kinetic symmetry breaking expands the survival range of $V$ (Fig. 2a, b). Kinetic symmetry breaking biases a copy-number ratio between complementary strands. This bias, like sex-ratio bias, must decrease the effective population size of replicators within a protocell. This inference can be formalised as follows. Consider a finite population of replicators with discrete generations. For simplicity, a population is assumed to consist entirely of either the plus strand or minus strand, alternating the sign in each generation (even if both strands initially coexist, one will eventually dominate owing to genetic drift, provided both are catalytic). The population size of each strand (denoted by $N_P$ and $N_M$) is assumed to be constant across generations. According to the population genetics theory[12], the variance effective population size (denoted by $N_e$) is approximated by the harmonic mean $2(1/N_P + 1/N_M)^{-1}$, which can be transformed into

$$N_e \approx (1 + \theta)(1 - \theta)\frac{N_P + N_M}{2},$$

where $\theta$ is the degree of copy-number asymmetry defined as $(N_P - N_M)/(N_P + N_M)$. If $N_P + N_M$ is kept constant, $N_e$ decreases with $|\theta|$. In other words, kinetic symmetry breaking increases genetic drift within a protocell.

To investigate the consequence of increased genetic drift, we considered the dynamics of a protocell along its line of descent. When a single protocell is continuously followed, it eventually ceases to reproduce because molecular-level evolution minimises $k_{xy}$ values (see the previous subsection). After ceasing to reproduce, the protocell starts to lose its replicators as they decay into diffusive substrates. Usually, the protocell dies with all its replicators decaying into substrates. Occasionally, however, it

resumes reproducing if, by chance, replicators with high-$k_{xy}$ values survive through genetic drift induced by a severe population bottleneck (Supplementary Fig. 5b). This restoration of growth is crucial for the survival of protocells, as attested by the fact that its prevention reduces the survival range of $V$ by five fold (Supplementary Fig. 6)[11].

Given the above dynamics of a protocell, we hypothesised that kinetic symmetry breaking increases the probability of bottleneck-induced growth restoration by decreasing the effective population size of replicators within a protocell. To test this hypothesis, we measured the probability of growth restoration as a function of the degree of kinetic asymmetry, defined as $\kappa \equiv (k_{PM} - k_{PP})/(k_{PM} + k_{PP})$ (see Methods section under 'The measurement of the probability of growth restoration'). The results show that the probability reaches a maximum value as $\kappa \to 1$ (Fig. 6a), supporting the above hypothesis (note, however, that $\kappa < 1$ is required for replicators to survive). To test the hypothesis further, we next measured the minimum number of replicators in a protocell that successfully restored its growth. The results show that this number tends to increase with $\kappa$ (Fig. 6b). In other words, kinetic asymmetry (i.e., high $\kappa$) enables a protocell to remove low-$k_{xy}$ replicators even when its internal replicators are still relatively abundant. This result suggests that kinetic asymmetry increases intracellular genetic drift. Taken together, the above results support the hypothesis that kinetic symmetry breaking increases the probability of bottleneck-induced growth restoration because it increases genetic drift within a protocell.

The increased probability of bottleneck-induced growth restoration explains why kinetic symmetry breaking expands the survival range of $V$ (Fig. 2). Note, however, that this is not the reason kinetic symmetry breaking occurs. Rather, kinetic symmetry breaking is a direct consequence of functional symmetry breaking, which creates selection pressure for an increased copy number of catalysts (Fig. 2c). Indeed, preventing bottleneck-induced growth restoration altogether does not prevent kinetic symmetry breaking (Supplementary Fig. 6).

**Sufficient conditions for functional symmetry breaking**. This section describes the results of our additional modelling to search for the minimal conditions required for functional symmetry breaking.

Above, functional symmetry breaking is ascribed to the deceleration of molecular-level evolution near the phase-space boundaries (see the section under 'The mechanism of the functional symmetry breaking'). Based on that argument, we hypothesised that the sufficient conditions for functional symmetry breaking are threefold. First, the fitness of replicators $f$ is a function of two mutationally-independent factors (denoted by $k_1$, $k_2$). Second, replicators are subject to conflicting multilevel evolution: molecular-level evolution tends towards minimising both $k_1$ and $k_2$; cellular-level evolution, towards maximising both $k_1$ and $k_2$. This condition implies that molecular- and cellular-level evolution are, each considered in itself, neutral with respect to the degree of asymmetry between $k_1$ and $k_2$. Third, the amount of variation in $f(k_1, k_2)$ due to mutation, $\sigma_f^2$, decreases as either $k_1$ or $k_2$ decreases. This condition can be fulfilled in various ways. For example, it suffices to assume that $\partial f(k_1, k_2)/\partial k_i = 0$ when $k_i < 0$.

To test the above hypothesis, we constructed a minimal model satisfying the above conditions by abstracting away all the model features that were considered unnecessary for symmetry breaking (Supplementary Methods under 'The minimal model displaying symmetry breaking'). Briefly, the model was formulated as a hierarchical Moran process[13]. Replicators are partitioned into protocells that divide when the number of internal replicators

exceeds $V$ (substrates were ignored). In each time step, one replicator replicates with a probability proportional to its fitness (complementarity and complex formation were ignored), and one replicator is removed randomly. The fitness of a replicator is defined as

$$f = \exp\left(\bar{k}_1 + \bar{k}_2 - r(k_1 + k_2)\right), \qquad (2)$$

where $k_1$ and $k_2$ are numerical values assigned to each replicator indicating the cooperativeness of the replicator (not the rates of complex formation anymore), $\bar{k}_1$ and $\bar{k}_2$ are the values averaged over all replicators in the same protocell, and $r$ is a constant indicating the relative cost of cooperation ($0 < r < 1$). The investigation of the above model shows that symmetry breaking occurs for an intermediate range of $V$, whereby the degree of asymmetry $|(k_1 - k_2)/(k_1 + k_2)|$ increases to unity, if $\sigma_f^2$ decreases as either $k_1$ or $k_2$ decreases (Supplementary Figs. 7 and 8). Moreover, symmetry breaking does not occur if $\sigma_f^2$ is invariant with respect to $k_1$ and $k_2$ (Supplementary Fig. 9). Taken together, these results suggest that the conditions described above are sufficient for symmetry breaking and that symmetry breaking is robust to the specific details of models.

**RNA folding genotype–phenotype map**. Although the conditions described in the previous section have the virtue of simplicity, the possibility of their fulfilment in reality is obscured by the fact that the phenotype (catalytic activity) of a replicator is a complex function of its genotype (base sequence). To address this issue, we extended the above minimal model by incorporating a complex genotype–phenotype map based on an RNA folding algorithm[14] (Supplementary Methods under 'The model incorporating the RNA folding genotype–phenotype map'). Briefly, each replicator is assigned a pair of complementary RNA sequences (genotype). These sequences determine the values of $k_1$ and $k_2$, which are defined as functions of the distances between the sequences' secondary structures, predicted by the folding algorithm[14], and an arbitrarily-chosen target structure. The investigation of this model shows that symmetry breaking occurs for an intermediate range of $V$ (Supplementary Figs. 10 and 11), suggesting that the conditions required for symmetry breaking can be fulfilled under a more realistic condition.

## Discussion
Functional and kinetic asymmetry between complementary strands has been suggested as the first, primordial form of differentiation between templates and catalysts in an RNA world[1]. To investigate the evolutionary mechanism and biological function of this asymmetry, we analysed a simple model of protocells in which replicating molecules undergo conflicting multilevel evolution. Evolution at the cellular level tends towards maximising the catalytic activities of the molecules, fostering cooperation among them. Conversely, evolution at the molecular level tends towards minimising their catalytic activities, breeding mutual exploitation within them. The results presented above indicate that these conflicting tendencies of evolution at multiple levels can induce symmetry breaking, whereby one strand of the molecules remains catalytic and increases its copy number (like enzymes), whereas the other strand becomes non-catalytic and decreases its copy number (like a genome). Importantly, these conflicting tendencies of evolution are directed along the axis of cooperation vs. exploitation, either maximising or minimising the catalytic activities of both strands of the molecules. Therefore, each of these tendencies alone is incapable of causing the asymmetry. When combined, however, they bring a new dimension to the evolution—i.e., the symmetry breaking— that is orthogonal to the prebuilt dichotomy between cooperation

and exploitation. Similar results have been obtained by previous studies investigating different models of multilevel evolution[15, 16]. Taken together, these results suggest that conflicting multilevel evolution can not only explain the primordial differentiation between templates and catalysts, but also, more generally, open up a new dimension in the evolutionary dynamics of hierarchically-organised replicating systems.

In the same vein, the symmetry breaking described above differs from the symmetry breaking in thermodynamic phase transitions. The former involves two tendencies, each of which cannot cause asymmetry alone (i.e., minimising or maximising $k_{xy}$). The latter also involves two tendencies, but one causes symmetry (entropy maximisation), whereas the other causes asymmetry (energy minimisation). The two cases differ in terms of whether it involves a tendency directly responsible for symmetry breaking.

The analysis of the model presented above indicates that the symmetry breaking can be analysed into the two aspects: functional and kinetic. The functional aspect refers to the differentiation between complementary strands in their catalytic activity (creating catalysts and templates); the kinetic aspect refers to the differentiation between complementary strands in their replication rates (creating majority and minority). The former causes the latter because it creates selection pressure to increase the relative copy number of catalysts per protocell[1, 3].

The functional symmetry breaking buffers a protocell against molecular-level evolution because it reduces the amount of variation generated by mutation in molecules within a protocell. Therefore, if molecular-level evolution is sufficiently rapid, the functional symmetry breaking ensues, increasing the equilibrium mean fitness of protocells. Conversely, if molecular-level evolution is too slow, the functional symmetry breaking does not occur because cellular-level evolution maximises the catalytic activity of both the strands of molecules.

The mechanism of the functional symmetry breaking summarised above is similar to that of the evolution of mutational robustness in quasispecies-like models[17–21]. Mutational robustness is the extent to which a system is buffered against deleterious mutations. If mutation pressure is sufficiently strong relative to selection pressure, mutational robustness evolves, increasing the equilibrium mean fitness of a population. Conversely, if mutation pressure is too weak, mutational robustness does not evolve because selection pressure maximises the fitness of individuals irrespective of mutational robustness (see the references cited above). As is clear from the comparison between this and the previous paragraphs, the evolution of mutational robustness is related to deleterious mutation in the manner in which functional symmetry breaking is related to molecular-level evolution. However, one difference is worth noting. Mutation in quasispecies-like models is random, whereas molecular-level evolution in the protocell model can switch between being random and non-random (i.e., neutral and non-neutral) depending on the population size of molecules within a protocell. This has a significant consequence for the dynamics of evolution as explored elsewhere[11]

Functional and kinetic symmetry breaking together generate non-catalytic, small-copy-number templates—the genome-like molecules. The paucity of the genome-like molecules increases genetic drift within a protocell and thereby further buffers protocells against molecular-level evolution. Recently, Bansho et al. have experimentally demonstrated that the number of templates per protocell must be sufficiently small for protocells to be evolutionarily stable[22, 23]. Taken together, these results suggest that small-copy-number templates provide long-term stability to the genetic information of protocells by preventing the evolution of selfish replicators within a protocell.

Previous theoretical studies have also indicated benefits of small-copy-number templates[24–27]. However, the mechanism by which such templates emerge has remained elusive. The present work takes a step towards addressing this issue by demonstrating the possibility that such templates emerge through spontaneous symmetry breaking between complementary strands induced by conflicting multilevel evolution.

Boza et al.[3] have investigated a model in which functional asymmetry between complementary strands is assumed a priori. Using this model, they have shown that functional asymmetry can induce kinetic asymmetry in the direction towards increasing the number of catalysts, a result partially overlapping with those presented in the present study (see Supplementary Fig. 12 for an explicit demonstration of the consistency with the previous model regarding this result). Going beyond this result, the present study presents two further results. First, functional asymmetry can be induced by conflicting multilevel evolution, so that it need not necessarily be regarded as a constraint of chemistry (see also the possibility of experimental tests described below). Second, kinetic symmetry breaking has a functionally significant consequence to cellular heredity because it generates genome-like molecules, which stabilise the genetic information of protocells.

The stabilisation caused by the paucity of the genome-like molecules might be compared to that caused by developmental bottlenecks in multicellular organisms and eusocial colonies. A multicellular organism typically develops from a single fertilised egg, and a eusocial colony often starts with a single mated queen. The paucity of elements (cells or organisms) establishing a new group (organism or colony) is thought to reduce conflicts of reproductive interest between elements constituting the group[28]. There is thus a parallel between the developmental bottleneck and genome-like molecules: in both cases, the paucity of elements from which a group derives impedes the evolution of selfish elements within that group. However, there is also a subtle difference. According to the kin selection theory, a developmental bottleneck is important because it increases relatedness within a group. By contrast, the paucity of the genome-like molecules stabilises protocells because it increases genetic drift within a protocell.

Note, however, that the paucity of the genome-like molecules is not an adaptation towards stabilising protocells against molecular-level evolution, but is a side effect of an adaptation towards increasing the copy number of catalysts per protocell (see refs [29, 30] for a potential parallel). Note also that the above comparison is not concerned with the evolution of a reproductive division of labour such as observed in multicellular organisms and eusocial colonies; rather, they focus on the paucity of elements from which a group derives. In fact, the symmetry breaking between complementary strands does not constitute a reproductive division of labour because both strands must serve as templates to complete the cycle of replication. However, a reproductive division of labour might also be induced by conflicting multilevel evolution, a possibility we are currently investigating. A relevant study has been published elsewhere[16].

To test the results described above, experiments must satisfy three conditions, which are the premises of the present work:

1. Replicating molecules are compartmentalised;
2. molecular-level evolution tends towards catalytic exploitation; conversely, cellular-level evolution tends towards catalytic cooperation and;
3. selection is exerted on both the strands of molecules.

The last condition is not satisfied by the current standard procedure for in vitro RNA evolution, in which only one strand is transcribed from dsDNA intermediates (e.g., ref. [31]). Consequently, ribozymes having catalytic activities in both strands are barely known to exist, with a notable exception[4]. Although such ribozymes are likely to be harder to synthesise owing to greater structural requirements[3], their possibility has not been experimentally explored yet. Finally, note also that experiments need not involve self-replicating RNA polymerase ribozymes (as assumed in the model), whose synthesis is still on the way[32–38] (but see below).

Recently, the synthesis of self-replicating RNA polymerase ribozymes entered on a new phase thanks to an innovation exploiting molecular chirality[39]. A major obstacle to this synthesis has been the severe template specificity of ribozymes due to the excessive dependence of template recognition on Watson–Crick pairing. Sczepanski and Joyce took a step towards addressing this issue by selecting a cross-chiral ribozyme that recognises its stereoisomer as a template, thus necessarily avoiding Watson–Crick pairing[39]. This experimental advance prompted us to investigate whether replicators envisaged in the Sczepanski–Joyce experiment would also undergo symmetry breaking similar to that reported above. To this end, we extended the model by incorporating a two-member hypercycle with complementary replication[40]. Briefly, the model assumes four classes of replicators: $L_P$, $L_M$, $D_P$, $D_M$. L and D denote different members of the hypercycle (stereoisomers); P and M denote complementary strands. L can catalyse the replication of D, but cannot catalyse that of L, and vice versa. Each replicator is assigned four complex formation rates $k_{xy}$ as before. For simplicity, L and D compete for the same substrates. The above model produced qualitatively the same results as presented above (Supplementary Fig. 13), indicating robustness and a greater potential for experimental tests of the present work.

## Methods

**The implementation of the model.** The model of Takeuchi et al.[11] was extended by incorporating complementary replication. One time step of the model consists of three sub-steps: the reaction, diffusion, and cell-division steps.

In the reaction step, the following algorithm is iterated $N/\alpha$ times, where $N$ is the number of particles, and $\alpha$ a scaling constant[5]. First, one particle (denoted by $X$) is chosen randomly from $N$ particles with an equal probability. Next, a second particle (denoted by $Y$) is chosen randomly from the protocell containing $X$ ($X$ and the molecule with which $X$ is forming a complex, if any, are excluded from this choice; the volume of a protocell is assumed to be proportional to the number of the particles in the protocell). If the protocell contains no molecule that can be chosen as $Y$, reactions involving $Y$ are excluded from the following algorithm. Depending on $X$ and $Y$, three types of reactions can occur:

1. If both $X$ and $Y$ are replicators that are not currently forming any complexes, they can form a complex. Two kinds of complexes are possible depending on which replicator serves as a catalyst or template. Complex formation in which $X$ serves as a catalyst and $Y$ as a template occurs with a probability $\alpha\beta k_{xy}^X$, where $k_{xy}^X$ is the complex formation rate of $X$, and $x$ and $y$ denote the strand types (P or M) of $X$ and $Y$, respectively ($\beta$ is described below). Complex formation in which $Y$ serves as a catalyst and $X$ as a template occurs with a probability $\alpha\beta k_{yx}^Y$.
2. If either $X$ or $Y$ is a replicator that is currently forming a complex and if the other is a substrate, replication occurs with a probability $\alpha\gamma$ ($\gamma$ is described below). Replication converts a substrate into a copy of the replicator serving as a template with possible mutation (see the next section).
3. If $X$ is a replicator, $X$ decays into a substrate with a probability $\alpha d$. If $X$ is forming a complex, the complex is dissociated before the decay. ($Y$ does not decay.)

Only one of the above reactions occurs with the given probability. To ensure that the relative frequencies of these reactions are proportional to their rate constants (namely, $k_{xy}^X$, $k_{yx}^Y$, 1, and $d$), the values of $\alpha$, $\beta$, and $\gamma$ were chosen as follows. The value of $\alpha$ was set such that the sum of the above probabilities never exceeded unity: $\alpha\left(\beta k_{xy}^X + \beta k_{yx}^Y + \gamma + d\right) \leq 1$. The value of $\beta$ was set to 1/2 in order to cancel out the fact that there are two possible orders in which a pair of particles are chosen to react. Likewise, $\gamma$ was set to 1/4 in order to cancel out the facts that there are two possible orders in which a complex and a substrate can be chosen (either the complex is chosen first or the substrate is chosen first) and that a complex has twice the chance of being chosen (a complex was considered to consist of two particles, each of which can be chosen independently). The above reaction algorithm was iterated $N/\alpha$ times per reaction step so that the time was independent of $\alpha$ and $N$. In this setting, any replicator has approximately a probability $d$ of decaying into a substrate per reaction step. Likewise, a replicator

(denoted by $X$) has approximately a probability $k_{xy}^X N_y^\nu / N^\nu$ of forming a complex in which $X$ serves as a catalyst, where $N_y^\nu$ is the number of the uncomplexed $y$-strand replicators in the protocell containing $X$, and $N^\nu$ is the number of the particles in the same protocell (note that $N^\nu$ was assumed to be proportional to the volume of the protocell). The above algorithm produces essentially the same dynamics as that of the Gillespie algorithm[41] if molecules are not partitioned into protocells[5].

In the diffusion step, all substrates are randomly re-distributed among protocells with probabilities proportional to the number of replicators in each protocell (the number of substrates within a protocell thus follows a multinomial distribution).

In the cell-division step, every protocell that has $\geq V$ particles is divided as described above.

Each simulation was run for at least $10^7$ time steps, unless otherwise stated. The value of $N$ was set to $50V$ so that the number of protocells was ~100 irrespective of the value of $V$, unless otherwise stated.

**The mutation of $k_{xy}$ values.** Mutation was modelled by three different methods in order to check the robustness of the results. In the first method (default), each $k_{xy}$ value was mutated by adding a number randomly drawn from a uniform distribution on the interval $(-\delta, \delta)$ ($\delta = 0.05$, unless otherwise stated). The values of $k_{xy}$ were bounded above with a reflecting boundary ($k_{xy} \leq 1$, unless otherwise stated), but were not bounded below in order to remove the boundary effect at $k_{xy} = 0$. When, however, $k_{xy} < 0$, the rate of complex formation was regarded as zero. The second method is nearly the same as the first, except that the boundary condition at $k_{xy} = 0$ was set to reflecting. In the third method, each $k_{xy}$ value was mutated by multiplying a number randomly drawn from a uniform distribution on the interval $(1 - \delta, 1 + \delta)$ (i.e., a random walk in a logarithmic scale). All three methods produce qualitatively identical results (Fig. 2 and Supplementary Fig. 1). Thus, for brevity, the results obtained with the first method are presented in Results, unless otherwise stated.

In Results (under 'The mechanism of the functional symmetry breaking'), it is argued that functional symmetry breaking decelerates molecular-level evolution because the amount of variation in the fitness of replicators within a protocell— denoted by $\sigma_f^2$ in Eq. (1)—decreases as $k_M \to 0$ (or $k_P \to 0$). This argument holds irrespective of the methods by which $k_{xy}$ values are mutated. In the first (default) method, $\sigma_f^2$ decreases, because $\partial f / \partial k_M = 0$ if $k_M < 0$. When $k_{xy}$ is mutated in a logarithmic scale (the third method), $\sigma_f^2$ decreases if $k_M \ll 1$, because $\sigma_{k_M}^2 \approx 0$ owing to vanishingly small mutation steps. The situation is similar when $k_{xy}$ has a reflecting boundary at $k_{xy} = 0$ (the second method), in which case $\sigma_{k_M}^2$ decreases because mutation can only increase $k_M$ when $k_M \approx 0$. (See Supplementary Methods under 'The minimal model displaying symmetry breaking' for a more mathematical description of the above argument in terms of the minimal model.)

**The correction for mutation step-size distributions.** Imposing functional or kinetic symmetry on the model causes correlations between $k_{xy}$ values. These correlations increase the amount of variation generated by mutation in the total catalytic activity of a replicator $\sum_{x,y} k_{xy}$. Increasing variation is similar to increasing the mutation rate; thus, it reduces the range of $V$ within which protocells survive. This confounding effect would obscure comparison between the models with and without symmetry imposed.

To remedy this issue, mutation step-size distributions were modified so as to keep the amount of variation generated by mutation in $\sum_{x,y} k_{xy}$ constant. Specifically, if functional symmetry is imposed (i.e., $k_{PP} = k_{MP}$ and $k_{PM} = k_{MM}$), $k_{PP}$ and $k_{MP}$ are mutated by adding $\sum_{i=1}^2 \epsilon_i / 2$, where $\epsilon_i$ is drawn independently from a uniform distribution on the interval $(-\delta, \delta)$ (the same applies to $k_{PM}$ and $k_{MM}$). If both functional and kinetic symmetry are imposed, each $k_{xy}$ value is mutated by adding $\sum_{i=1}^4 \epsilon_i / 4$.

Note that the survival ranges of $V$ are identical between the model in which functional symmetry is imposed (Fig. 2c) and the model in which both functional and kinetic symmetry are imposed (Fig. 2d). The equality of the survival ranges suggests that the confounding effect mentioned above is successfully removed. Because the two models display no symmetry breaking (Fig. 2c, d), they are expected to display an identical survival range of $V$ if imposing symmetry causes no additional effects besides preventing symmetry breaking.

**The measurement of the number of cell divisions.** To observe the dynamics of a single protocell along its line of descent, the protocell was assumed to compete against a very large, stationary population of protocells (hereafter, referred to as background population). Let $N_R'$ and $N_S'$ be the total number of replicators and substrates, respectively, in the background population. Let $n_R$ and $n_S$ be the number of replicators and substrates, respectively, in the protocell under observation. To dispense with the computation of the background population, it was assumed that $N_S' + N_R' \to \infty$ with $N_S'/N_R'$ kept constant. Under this assumption, the model was modified as follows. In the diffusion step, $n_S$ was set to a number drawn from a Poisson distribution with a mean $n_R N_S'/N_R'$. The reaction step was not modified. In the cell-division step, if the protocell divided, one of the daughter cells was randomly chosen and discarded, and the other was continuously followed. At the beginning of each simulation, a protocell was initialised with a homogeneous population of replicators. The $k_{xy}$ values of those replicators were chosen such that the expected density of replicators (denoted by $E_0[n_R/(n_R + n_S)]$) was constant in

the absence of substrate diffusion across protocells. This density was set greater than $N_R'/(N_R' + N_S')$ so that a protocell under observation was always initially at a competitive advantage relative to the background population (specifically, $E_0[n_R/(n_R + n_S)] = 0.725$; $N_R'/(N_R' + N_S') = 0.7$). For simplicity, kinetic symmetry was imposed on the model, as described in the previous section. Simulations were repeated at least $10^4$ times for each set of initial $k_{xy}$ values, where $d$ was set to 0.0632, $V$ was set to 1700, and $k_{xy}$ was not bounded above.

**The measurement of the probability of growth restoration.** The same types of simulations as described in the previous section was performed using the following parameters: $E_0[n_R/(n_R + n_S)] = 0.88$, $N_R'/(N_R' + N_S') = 0.85$, $d = 0.02$, $V = 2500$. Bottleneck-induced growth restoration was defined as the event in which a protocell undergoes at least one cell division after the number of its internal particles has decreased below $0.3 V$. Simulations were made with the model in which no symmetry was imposed, and $k_{xy}$ was not bounded above. Simulations were repeated at least $10^5$ times for each set of initial $k_{xy}$ values. The minus strand was assumed to be non-catalytic: $k_{MP}$ and $k_{MM}$ were set to a negative value so that they remained irrelevant during simulations.

**Code availability.** The C++ source codes implementing the models are available from the corresponding author upon request.

**Data availability.** The authors declare that the data supporting the findings of this study are available within the paper and its Supplementary Information files.

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

## Acknowledgements

N.T. and K.K. are supported by the Dynamic Approaches to the Living Systems from MEXT, Japan. P.H. is supported by EU FP7 EVOEVO project (ICT-610427). The authors thank Daniel van der Post for comments on the manuscript. N.T. is supported by JSPS KAKENHI Grant Number JP17K17657.

## Author contributions

N.T. conceived the study, designed, implemented and analysed the model, and wrote the paper. K.K. and P.H. discussed the design, results and implications of the study, and commented on the manuscript at all stages.

## Additional information

**Competing interests:** The authors declare no competing financial interests.

