## [Peer Review file · Nature Communications]

Reviewers' comments:

Reviewer #1 (Remarks to the Author):

Review: The origin of genes through spontaneous symmetry breaking
Authors: N. Takeuchi, P. Hogeweg and K. Kaneko

General comments

Overall I enjoyed reading this submission and thought that, despite the purely theoretical nature of the paper, it makes an important contribution to our understanding of the separation of genotype and phenotype and genetic complexity at the origin of life. Specifically the evidence that conflict between the two levels of selection drove what they call "symmetry breaking" is original and is likely to direct a new line of enquiry into the origin of genetic complexity. The theoretical model is very attractive but what weakens it is that the language used is not rooted in empirical biology. The language is very confusing and requires clarity. That is the main weakness of this submission and explained below, although I do think this can be overcome by rewriting sections with more appropriate language. I also think that some of the referencing can be improved or more appropriate to strengthen the arguments and assumptions made in the model.

Specific comments

1.

The gene concept is fraught with various issues and the authors use it inconsistently (see for example the paper by Griffiths Bioscience, 1999, "The many faces of the gene"). I had to read the article several times before settling on what I thought the authors mean so I think it would be helpful to state a priori what is meant by "gene". Sometimes the authors seem to use it as the "molecular biology gene", other times they use it in the context of the "evolutionary gene concept". Perhaps they have their own interpretation but either way it would be helpful to know exactly what is being referred to. The reason is that it has implications for interpreting the theoretical model results. And of course, it is helpful to clarify which gene they are referring to when speaking about the unit of selection at the molecular level in the model.

There is another point to make about the use of the term gene. I am assuming that the genes here are not protein coding (the genetic code and proteins have not yet evolved in the RNA world which the authors refer to)? I think it's important to make this explicit for the benefit of molecular biologists, evolutionary biologists or theoreticians all of whom have different default interpretations of what is meant by "gene". Eg line 60, p2 the authors say "(like genes)". These are not genes as we conventionally use the term in either molecular biology or evolutionary biology. Another example of the inconsistent terminology can be seen in the first few lines of the abstract. The authors first speak about non-catalytic templates, the genes, referring specifically to the genes in a cell. In the next sentence they speak about gene-like molecules arising from symmetry breaking. While the whole concept and study I think is very interesting the terminology is confusing. The first usage (genes in the cell) are explained using a second, entirely different concept of gene-like molecules. The first have a genetic code, separation of genotype and phenotype, promoters and regulatory elements. The second I am assuming are non-coding RNA templates.

2.

It is true that both complementary strands could have catalytic activity. As the authors indicate (reference 4) having catalytic activity in both strands would be far more efficient. In reality this is very rare. Of course that doesn't mean it didn't happen at the origin of life but it is implied that activity in both strands is always possible, which is seldom the case in more complex organisms. In viruses and less complex prokaryotes this phenomenon is more common and perhaps the authors should indicate that.

3.

Each replicator is assumed to be capable of serving as both a catalyst and a template. This is true; however, as the authors state this cannot be at the same time. I think this constraint must be made more explicit. The reason is that many theoreticians and origin of life biochemists speak about "replicases", which would lead them to question the model in its entirety. In fact whole research programs are developed to manufacture replicases! But I agree with the authors here because replicases probably never existed and this should be explained. A template cannot be unwound (linear to act as a template) and wound (to act as a catalyst) at the same time. I think the authors need to refer to some of the works that explain why molecules cannot serve as catalyst and template simultaneously (Durand and Michod, 2010, Evolution "Genomics in the light of evolutionary transitions"; Lincoln and Joyce, 2009, Science "Self-sustained replication of an RNA enzyme").

4.

The explanation for selection at the level of protocells is not explained well. I think it should be made explicit how protocells may compete against each other and what criteria would need to be in place for that to happen. A number of assumptions are being made in the model at this point and I think explaining or mentioning them would be helpful.

5.

The point about conflict at the two levels is well thought out and quite compelling. I think; however, that this conflict as the proven cause for symmetry breaking does not automatically follow. It is a viable explanation but it is not proved. I suggest leaving out "therefore", line 161, page 5.

6.

The paragraph p11, lines 424-437, seems to be unjustified. I understand that the authors may wish to highlight similarities in other systems like multicellularity and eusocial communities of insects, but in my view the comparison is neither valid nor necessary. It actually weakens their paper, which is strong enough and doesn't need this tangential comparison. I think the authors wish to show that the findings of their study have parallels in other examples where multilevel selection plays a role, but it doesn't seem appropriate and detracts from their own story because the parallels are weak.

7.

Line 451, p12 should read: "Selection is exerted on both the strands of molecules."

8.

Lines 22-24, p 1. The statement that in modern cells all templates produce catalysts is simply not true. Most templates produce structural inert proteins.

9.

The mathematical model is explained well and the figures and graphs explain the results beautifully. The only comment I have is that the assumptions should be explained better, particularly with regards selection between protocells (see comment 4).

Reviewer #2 (Remarks to the Author):

In this model the number of strands N is fixed. When one reproduces, another dies. It would also be possible to keep the number of cells fixed. Replicate the strands as in the current model, then divide a cell when it reaches V strands, then kill another cell when the first one divides. What kind of resource limitation keeps the population fixed in this model, and why is it the strands that are

fixed rather than the cells?

Is this model well-defined in the limit of N goes to infinity? You could have the number of strands and the number of cells infinite, but keep V finite. I think it would still work. That means N is not a very important parameter. Everything depends on V .

The model assumes that each strand is either a P or an M, but nevertheless it possesses four k properties. Only two of these are expressed phenotypes of the strand, but the other two get expressed at the next generation, so it is necessary for both P and M strands to have all four properties associated with them. This makes sense to me, but perhaps it could be stated a little more clearly at the beginning.

There is an important assumption hidden in the model definition. The four k values are all properties of the catalyst and not properties of the template. In the model description (line 481) it is the k of strand X that matters. The only thing about Y that matters is that it is a P or an M. It is not obvious to me that all P or all M strands should be treated equally by any one catalyst. It would be possible to define a model where the rate depended on the catalytic ability of X and the template ability of Y.

There are not really any parasites in this model. Even in cases where the k 's go to zero on the M strand, the M strand still retains the coding ability for the functional P strand. Shouldn't mutations simply create non-functional parasitic templates that have no coding ability for the k 's any more.

It is assumed from the beginning that it is possible for both P and M strands to be equally functional at the same time. This is tested briefly in the section with the RNA folding model. It seems to me unlikely that an M strand will have the same structure as the P strand. I would guess that there is an extremely small phenotypic error threshold for such symmetric P/M sequences, whereas if only the P strand needs to be functional, the phenotypic error threshold will be much higher. I doubt that you could ever have a ribozyme of large size and high catalytic rates for which both strands are functional (maybe the Lincoln and Joyce example of the ligase is a counter example - but surely this is a very special case).

Under this assumption, the main model (e.g. in Fig 2) should have $k_{MM} = k_{MP} = 0$ always, but begin with $k_{PP} = k_{PM} = 1$. This would be imposed by the nature of RNA. Then you would presumably evolve kinetic symmetry breaking where $k_{PM} > k_{PP}$ so that you increase the number of P relative to M strands. Could this case be added to figure 2? It seems to be the most important case to me and more realistic than the other four cases.

The sentence 'Without loss of generality...' at the bottom of p7 cannot be true if there is complete functional symmetry breaking, because the population would die completely if it was all non-functional M strands.

Fig 2d is interesting. There are two stable states for a short range around $V = 700-800$. This looks like a first order jump. It says the white points are metastable - but they should be stable for a very long time I think? Why are there white points at the top and the bottom of this jump? At least one of these is stable. I would guess both are stable for large populations.

Similarly 2b is interesting. There looks like a jump where k_{MM} goes to zero, but then there is a region with high k_{PP} , followed by a second jump. Should there be some white points at the bottom of the second jump?

I was confused by the supplementary section 1.1 with the minimal model because this does not describe the main model in the text, and because I read this section before I realized that there is another methods section at the end of the main paper.

In the supplementary methods, \bar{k} is an average over all replicators in the same cell. Shouldn't it be an average over all the other replicators except the template strand in question? In this same model, the fitness does not depend on the number of replicators in one cell, but only on the k -

bars. Shouldn't the replication rate of a strand depend on the number of catalysts in the cell?
There is some assumption here about concentrations of strands in a cell. It seems to be assumed here that the concentration of strands stays fixed when the number of strands goes up. Maybe this is true if volume is proportional to the number of strands.

I don't think the model in supplementary section 1.1 adds to the paper very much, and given the potential confusion with the main model, it might be better to remove this section.

Is there any experimental information that tells us how big V should be in a protocell. Having $V =$ hundreds or thousands seems a lot to me. I had always imagined $V = 10$ or so.

Reply to Reviewer 1's comments:

We have revised the manuscript based on the reviewer's comments as described below. In this revised submission, we are providing two versions of an identical manuscript, with and without all revisions highlighted. The page and line numbers shown below are for the version with the highlighting, which is uploaded as Related Manuscript Files.

> General comments

> Overall I enjoyed reading this submission and thought that, despite
> the purely theoretical nature of the paper, it makes an important
> contribution to our understanding of the separation of genotype and
> phenotype and genetic complexity at the origin of life. Specifically
> the evidence that conflict between the two levels of selection drove
> what they call "symmetry breaking" is original and is likely to
> direct a new line of enquiry into the origin of genetic
> complexity. The theoretical model is very attractive but what weakens
> it is that the language used is not rooted in empirical biology. The
> language is very confusing and requires clarity. That is the main
> weakness of this submission and explained below, although I do think
> this can be overcome by rewriting sections with more appropriate
> language. I also think that some of the referencing can be improved or
> more appropriate to strengthen the arguments and assumptions made in
> the model.

We are encouraged by the reviewer's general comments. We slightly elaborated on the term "symmetry breaking" by adding the following text: "symmetry breaking (i.e. the average values of k_{xy} are not invariant under the exchange of P and M)" (lines 154-155, page 4).

> Specific comments

> 1. The gene concept is fraught with various issues and the authors use
> it inconsistently (see for example the paper by Griffiths Bioscience,
> 1999, "The many faces of the gene"). I had to read the article
> several times before settling on what I thought the authors mean so I
> think it would be helpful to state a priori what is meant by
> "gene". Sometimes the authors seem to use it as the "molecular
> biology gene", other times they use it in the context of the
> "evolutionary gene concept". Perhaps they have their own
> interpretation but either way it would be helpful to know exactly what
> is being referred to. The reason is that it has implications for
> interpreting the theoretical model results. And of course, it is
> helpful to clarify which gene they are referring to when speaking
> about the unit of selection at the molecular level in the model.

What we meant by the word "gene" in the original manuscript is molecules sharing the two features of "molecular biology genes": (1) they are non-catalytic templates providing information for producing catalysts; (2) their copy number is smaller than that of the catalysts encoded by them. As pointed out by the reviewer, our usage of the term needed more qualification. To overcome this weakness, we revised the manuscript based on the reviewer's next comment as described below.

> There is another point to make about the use of the term gene. I am
> assuming that the genes here are not protein coding (the genetic code
> and proteins have not yet evolved in the RNA world which the authors
> refer to)? I think it's important to make this explicit for the
> benefit of molecular biologists, evolutionary biologists or
> theoreticians all of whom have different default interpretations of
> what is meant by "gene". Eg line 60, p2 the authors say "(like
> genes)". These are not genes as we conventionally use the term in
> either molecular biology or evolutionary biology. Another example of
> the inconsistent terminology can be seen in the first few lines of the
> abstract. The authors first speak about non-catalytic templates, the
> genes, referring specifically to the genes in a cell. In the next
> sentence they speak about gene-like molecules arising from symmetry
> breaking. While the whole concept and study I think is very
> interesting the terminology is confusing. The first usage (genes in
> the cell) are explained using a second, entirely different concept of
> gene-like molecules. The first have a genetic code, separation of
> genotype and phenotype, promoters and regulatory elements. The second
> I am assuming are non-coding RNA templates.

The reviewer's comment, together with Griffiths & Neumann-Held (1999 Bioscience) cited therein, led us to realize that the word "gene" has many misleading implications in the context of the present study. The chief problem among these is that a "molecular biology gene" is always defined as a protein-coding unit and, therefore, necessarily implies translation, a process that is outside the scope of the present study, but nevertheless might appear relevant because the present study is concerned with differentiation between templates and catalysts. To correct this defect, we revised the manuscript in two ways.

First, we think that the word "gene" could be replaced with the less problematic word "genome" for the following reason. According to a standard textbook, the genome of a cell is "the totality of its genetic information as embodied in its complete DNA sequence" (Alberts et al. 2014 Molecular Biology of the Cell). Alternatively, a dictionary defines it as "the complete set of genes or genetic material present in a cell or organism" (<https://en.oxforddictionaries.com/definition/genome>). According to these definitions, a genome does not necessarily carry molecular biological implications that are misleading in the context of the present study. Therefore, "genome" seems more suitable than "gene". Accordingly, we have replaced "gene" with "genome" throughout the manuscript. However, this revision goes beyond what has been explicitly suggested by the reviewer (which we have also incorporated as described next). Thus, we would like to have the reviewer's opinion on our choice. If there are good reasons to revert to "gene", we will do so.

Second, in the revised manuscript, we have explicitly stated that by "genome-like molecules" we mean molecules having the two features of a genome as mentioned above (lines 37-39, page 2). Moreover, in Introduction, we outright state that the present study focuses on these two particular features shared by all genomes, implying that we are not

concerned with the concept of "molecular biology genome" in its entirety (lines 25-27, page 1).

> 2. It is true that both complementary strands could have catalytic
> activity. As the authors indicate (reference 4) having catalytic
> activity in both strands would be far more efficient. In reality this
> is very rare. Of course that doesn't mean it didn't happen at the
> origin of life but it is implied that activity in both strands is
> always possible, which is seldom the case in more complex
> organisms. In viruses and less complex prokaryotes this phenomenon is
> more common and perhaps the authors should indicate that.

At first sight, we thought the possession by extant viruses and prokaryotes of "overlapping genes", i.e. genes encoded by overlapping complementary strands of genomic DNA molecules, supports the assumption that both complementary strands can have catalytic activity and the inference that having catalytic activity in both strands would be more efficient. However, we found a few existing studies undermining this comparison. Specifically, Brandes & Linial (2016 Biol Direct 11:26) provides evidence suggesting that overlapping genes in viruses are by-products of high mutation pressure. This study suggests the possibility that many overlapping genes in viruses, which are typically short, might not be functional. Johnson & Chisholm (2004 Genome Res 14:2268-72) provides evidence suggesting that overlapping genes in prokaryotes have function in gene regulation, a process that is not considered in the above inference about the efficiency of ribozymes. In addition, Johnson & Chisholm (2004) shows that the fraction of overlapping genes is almost independent of the genome size.

Given the above considerations and also the fact that overlapping genes imply translation (comment 1), we chose not to refer to overlapping genes and, accordingly, removed the citation of Rodin & Ohno (1995) in the revised manuscript. Instead, we have added the elaboration of the above assumption and inference to support their reasonableness. First, we now state in Discussion that the current standard procedure for in vitro RNA evolution selects only one strand, so that the possibility of ribozymes having catalytic activities in both strands has not been experimentally explored yet (lines 527-530, page 13), implying that the rarity of such ribozymes currently known to us cannot be taken as strong evidence against the assumption. Second, we have added a text in parentheses to mention the reasoning behind the inference; namely, "Having catalytic activity in both strands would be far more efficient (particularly in the absence of transcriptional or translational amplification)" (lines 54-55, page 2).

> 3. Each replicator is assumed to be capable of serving as both a
> catalyst and a template. This is true; however, as the authors state
> this cannot be at the same time. I think this constraint must be made
> more explicit. The reason is that many theoreticians and origin of
> life biochemists speak about "replicases", which would lead them to
> question the model in its entirety. In fact whole research programs
> are developed to manufacture replicases! But I agree with the authors
> here because replicases probably never existed and this should be
> explained. A template cannot be unwound (linear to act as a template)

> and wound (to act as a catalyst) at the same time. I think the authors
> need to refer to some of the works that explain why molecules cannot
> serve as catalyst and template simultaneously (Durand and Michod,
> 2010, Evolution "Genomics in the light of evolutionary transitions";
> Lincoln and Joyce, 2009, Science "Self-sustained replication of an
> RNA enzyme").

In the revised manuscript, we have cited Durand and Michod (2010), which we had inadvertently overlooked, in Introduction, where the trade-off is first mentioned (line 63, page 2). We would like to note here that the trade-off in our model is not directly assumed by imposing a constraint on the allowable values of parameters. Rather, it emerges from the two-step replication that involves complex formation (Takeuchi & Hogeweg 2007 J Mol Evol). Therefore, we have also cited the latter work together with Durand and Michod (2010) in Introduction. Moreover, in Model, we have added that our assumption is based on the constraint, which is likely to exist in RNA molecules, that providing catalysis and serving as a template impose structurally-incompatible requirements, citing Durand and Michod (2010) again (lines 89-91, page 3).

> 4. The explanation for selection at the level of protocells is not
> explained well. I think it should be made explicit how protocells may
> compete against each other and what criteria would need to be in place
> for that to happen. A number of assumptions are being made in the
> model at this point and I think explaining or mentioning them would be
> helpful.

Selection between protocells occurs through competition for substrates, which are generated only through the decay of replicators and passively diffuse across protocells. To explain these important modeling assumptions, we have added a few sentences to the manuscript under "Model" (lines 125-131, page 4).

> 5. The point about conflict at the two levels is well thought out and
> quite compelling. I think; however, that this conflict as the proven
> cause for symmetry breaking does not automatically follow. It is a
> viable explanation but it is not proved. I suggest leaving out
> "therefore", line 161, page 5.

Based on this comment, we revised the manuscript in two ways. First, we have replaced "therefore" with "the above consideration suggests that." Second, we have added a paragraph, together with a new Supplementary Figure, describing a newly obtained result that confirms our inference (lines 203-211, pages 5-6; Supplementary Fig. 3). Specifically, we investigated a model that incorporates one-step replication (instead of two-step replication). This model assumes that replication is instantaneous and thereby abstracts away complex formation. In this model, no trade-off emerges between templates and catalysts, so that there is no conflicting multilevel evolution. The model displays no symmetry breaking, confirming that conflicting multilevel evolution is a key cause of the symmetry breaking.

> 6. The paragraph p11, lines 424-437, seems to be unjustified. I
> understand that the authors may wish to highlight similarities in

> other systems like multicellularity and eusocial communities of
> insects, but in my view the comparison is neither valid nor
> necessary. It actually weakens their paper, which is strong enough and
> doesn't need this tangential comparison. I think the authors wish to
> show that the findings of their study have parallels in other examples
> where multilevel selection plays a role, but it doesn't seem
> appropriate and detracts from their own story because the parallels
> are weak.

We have replaced a chief portion of the paragraph with a very short text in parentheses, "(see [30, 31] for a potential parallel)" (line 510, p 13). We have kept the first sentence of the paragraph, which states that the evolution of genome-like molecules is not an adaptation towards stabilizing protocells, but is a side effect of an adaptation towards increasing the copy number of catalysts per protocell (lines 496-498, pages 12). We consider that this point is worth making independently of the deleted comparison.

> 7. Line 451, p12 should read: "Selection is exerted on both the
> strands of molecules."

We have corrected this error (line 525, page 13, and line 445, page 11).

> 8. Lines 22-24, p 1. The statement that in modern cells all templates
> produce catalysts is simply not true. Most templates produce
> structural inert proteins.

We have inserted a text in Introduction to indicate that not all parts of genomic templates provide information for producing catalysts (line 28, page 1).

> 9. The mathematical model is explained well and the figures and graphs
> explain the results beautifully. The only comment I have is that the
> assumptions should be explained better, particularly with regards
> selection between protocells (see comment 4).

In addition to the revision based on the reviewer's comment 4, we have elaborated on our modeling assumptions based on Reviewer 2's comments.

Reply to Reviewer 2's comments:

We have revised the manuscript based on the reviewer's comments as described below. In this revised submission, we are providing two versions of an identical manuscript, with and without all revisions highlighted. The page and line numbers shown below are for the version with the highlighting, which is uploaded as Related Manuscript Files.

> In this model the number of strands N is fixed. When one reproduces,
> another dies. It would also be possible to keep the number of cells
> fixed. Replicate the strands as in the current model, then divide a
> cell when it reaches V strands, then kill another cell when the first

> one divides. What kind of resource limitation keeps the population
> fixed in this model, and why is it the strands that are fixed rather
> than the cells?

The parameter N is the total number of replicators and substrates (collectively called particles). A replicator consumes one substrate to replicate and generates one substrate by decaying, thereby keeping N constant. Thus, the total number of replicators is not strictly fixed, but fluctuates around a certain value at equilibrium. To improve the clarity of the model description, in the revised manuscript, we have more explicitly stated that what is kept constant is the total number of replicators and substrates (lines 79-80, page 3).

We chose the above method of modeling in order that selection at the molecular and cellular levels both emerge from an identical process, i.e. substrate competition. We are unaware of any reason to suspect that our conclusions depend on whether the total number of protocells is strictly kept constant or not.

> Is this model well-defined in the limit of N goes to infinity? You
> could have the number of strands and the number of cells infinite, but
> keep V finite. I think it would still work. That means N is not a very
> important parameter. Everything depends on V .

We think the model is well defined in the limit of N tending to infinity with V kept finite. The exact value of N is unimportant if N is sufficiently large. If, however, N is too small, cellular-level genetic drift would dominate over cellular-level selection, driving protocells to extinction. In our model, the number of protocells approximately tends to $2N/V$ (this information has been added to the manuscript under Model, lines 133-134, page 4). The value of $2N/V$ was set to 100.

> The model assumes that each strand is either a P or an M , but
> nevertheless it possesses four k properties. Only two of these are
> expressed phenotypes of the strand, but the other two get expressed at
> the next generation, so it is necessary for both P and M strands to
> have all four properties associated with them. This makes sense to me,
> but perhaps it could be stated a little more clearly at the beginning.

The additional clarification suggested by the reviewer is highly beneficial. In the revised manuscript, we have stated, "Among the four k_{xy} values of a given replicator, two values denote the rates at which this replicator, serving as a catalyst, forms a complex ... ; the other two values apply to the complement of this replicator" (lines 108-112, page 3).

> There is an important assumption hidden in the model definition. The
> four k values are all properties of the catalyst and not properties of
> the template. In the model description (line 481) it is the k of
> strand X that matters. The only thing about Y that matters is that it
> is a P or an M . It is not obvious to me that all P or all M strands
> should be treated equally by any one catalyst. It would be possible to
> define a model where the rate depended on the catalytic ability of X
> and the template ability of Y .

First, we would like to note that in our model, the trade-off between templates and catalysts emerges from the assumption that replication takes a finite amount of time, during which a catalyst cannot serve as a template. Within this modeling framework, we can conceive two ways to incorporate the template ability of Y. In the first way, the template ability of Y is simply added to the model as another k parameter. In this case, we expect that the template ability is always maximized because increasing this ability is beneficial both at the molecular and cellular levels. In the second way, the recognition of templates by catalysts is incorporated into the model by sequence matching as done in Takeuchi & Hogeweg (2008 *Biology Direct* "Evolution of complexity in RNA-like replicator systems"). In this case, the model becomes considerably more complex and, therefore, is beyond the scope of the present study.

> There are not really any parasites in this model. Even in cases where
> the k 's go to zero on the M strand, the M strand still retains the
> coding ability for the functional P strand. Shouldn't mutations simply
> create non-functional parasitic templates that have no coding ability
> for the k 's any more.

Prompted by this comment, we tested the effect of incorporating mutation that creates non-functional parasitic templates into the model. The results remained qualitatively the same as obtained with the original model. However, the critical value of V above which symmetry breaking occurs decreases to a value between 178 and 316. This decrease is expected, given that incorporating the above mutation is similar to increasing the mutation rate. In the revised manuscript, we have mentioned this result (lines 173-175, page 5) and added a new Supplementary Figure displaying it (Supplementary Figure 2)

> It is assumed from the beginning that it is possible for both P and M
> strands to be equally functional at the same time. This is tested
> briefly in the section with the RNA folding model. It seems to me
> unlikely that an M strand will have the same structure as the P
> strand. I would guess that there is an extremely small phenotypic
> error threshold for such symmetric P/M sequences, whereas if only the
> P strand needs to be functional, the phenotypic error threshold will
> be much higher. I doubt that you could ever have a ribozyme of large
> size and high catalytic rates for which both strands are functional
> (maybe the Lincoln and Joyce example of the ligase is a counter
> example - but surely this is a very special case). Under this
> assumption, the main model (e.g. in Fig 2) should have $k_{MM} = k_{MP} = 0$
> always, but begin with $k_{PP} = k_{PM} = 1$. This would be imposed by the
> nature of RNA. Then you would presumably evolve kinetic symmetry
> breaking where $k_{PM} > k_{PP}$ so that you increase the number of P relative
> to M strands. Could this case be added to figure 2? It seems to be the
> most important case to me and more realistic than the other four
> cases.

The reviewer's comment led us to examine the consequence of directly assuming functional asymmetry in the model (i.e. $k_{MM} = k_{MP} = 0$ throughout a simulation). As the reviewer expected, the result shows

that kinetic symmetry breaking occurs in the direction towards increasing the number of catalysts (i.e. $k_{PM} > k_{PP}$). In the revised manuscript, we have added a new Supplementary Figure that displays the above result (Supplementary Figure 12) together with the additional discussion as described below.

Although the above result is important with respect to the suggestion of Szathmary and Maynard Smith (1994 J Theor Biol) mentioned in Introduction, we chose not to make it the most important result of the present study for two reasons. First, this result has already been obtained by Boza et al. (2014 PLoS Comp Biol) cited in the manuscript. Specifically, Boza et al. have investigated the following question and gave an affirmative answer to it:

(1) Does functional asymmetry lead to kinetic asymmetry?

Second, our work, by contrast, specifically investigates the two questions stemming from the work of Szathmary and Maynard Smith (1994 J Theor Biol) that have not been addressed by Boza et al. Namely,

(2) How does functional asymmetry evolve?

(3) What is the genetic significance of kinetic asymmetry?

Question (1) bridges the gap between Questions (2) and (3). Thus, answering all these questions make a whole story. Given this consideration, we added in Discussion a new paragraph making the following two points (lines 474-484, page 12). First, we explicitly mention the result of Boza et al. and refer to the newly obtained result described above to indicate that our modeling outcome is consistent with that of Boza et al. Second, our work extends the earlier work by addressing the above two distinct questions.

Prompted by the reviewer's doubt about the possibility of ribozymes having catalytic activity in both strands, we have also revised Discussion by stating that the current standard procedure for in vitro RNA evolution selects only one strand, so that the possibility of ribozymes having catalytic activities in both strands has not been experimentally explored yet (lines 527-530, page 13), implying that the rarity of such ribozymes currently known to us cannot be taken as strong evidence against the assumption. We have also cited Lincoln and Joyce in Introduction.

> The sentence 'Without loss of generality...' at the bottom of p7
> cannot be true if there is complete functional symmetry breaking,
> because the population would die completely if it was all
> non-functional M strands.

We replaced the phrase "Without loss of generality" with "For simplicity" (line 321, page 8). We also elaborated the text in parentheses as follows: "even if both strands initially coexist, one will eventually dominate owing to genetic drift, provided both are catalytic" (line 324, page 8).

> Fig 2d is interesting. There are two stable states for a short range

> around $V = 700-800$. This looks like a first order jump. It says the
> white points are metastable - but they should be stable for a very
> long time I think? Why are there white points at the top and the
> bottom of this jump? At least one of these is stable. I would guess
> both are stable for large populations.

The metastable states represented by the white points refer to states that are reachable from different initial conditions with no state transition observed within $1e7$ time steps (a replicator decays approximately with a probability $d=0.02$ per time step). We added this important qualification to the revised manuscript (lines 810-812, page 23).

Indeed, the existence of multi-stability is interesting. It would be beneficial to see what would happen to these states if N tended to an extremely large value, as the reviewer suggested, but this is beyond the available computational capability. However, we have some preliminary results that we can share, which shed some light on the existence of multi-stability. In the minimal model described in Supplementary Methods, the fitness function is defined as an exponential function of the cooperativeness of replicators (i.e. a linear function of cooperativeness in the continuous-time limit). Therefore, the absolute value of the cooperativeness cannot affect the evolutionary dynamics, a condition excluding the possibility of multi-stability. However, when we examined a non-exponential fitness function (i.e. nonlinear fitness function in the continuous-time limit), we observed the existence of multi-stability in the minimal model, despite the fact that the fitness function is strictly monotonic with respect to the cooperativeness (data not shown). This result indicates that the existence of multi-stability has something to do with the interplay between multilevel selection and the curvature of a fitness function (i.e. the second derivative of the logarithm of a fitness function). The detailed exploration of this preliminary result seems worthwhile, but is clearly beyond the scope of the current study.

> Similarly 2b is interesting. There looks like a jump where k_{MM} goes to
> zero, but then there is a region with high k_{PP} , followed by a second
> jump. Should there be some white points at the bottom of the second
> jump?

There might indeed be such points that have not been detected owing to the resolution of our parameter search. If this question is to be pursued, though, the minimal model described in Supplementary Methods might be more fitting, not only because it is simpler, but also because the question is independent of the RNA world hypothesis (please see also below for why we kept the minimal model in the revised manuscript).

> I was confused by the supplementary section 1.1 with the minimal model
> because this does not describe the main model in the text, and because
> I read this section before I realized that there is another methods
> section at the end of the main paper.

The reviewer's comment made us realize that we should have minimized the possibility of such confusion. In the revised Supplementary Methods, we

outright state that the model described therein is concerned with the verification of the hypothesis regarding the sufficient conditions for symmetry breaking and is distinct from the main model. In addition, we have added therein a reference to the Methods section of the main text for the description of the main model.

> In the supplementary methods, k -bar is an average over all replicators
> in the same cell. Shouldn't it be an average over all the other
> replicators except the template strand in question?

In the minimal model, the variables, k_1 and k_2 , denote the cooperativeness of an individual and do not specifically refer to the rate of complex formation (we added this information in the revised manuscript; line 384-385, page 10). This is the reason why we did not define k -bar as an average excluding the template in question. In other words, our conclusion is independent of whether the altruism is "strong" or "weak" in D.S. Wilson's sense (this is one of the reasons we think we should keep the minimal model in the revised manuscript).

> In this same model, the fitness does not depend on the number of
> replicators in one cell, but only on the k -bars. Shouldn't the
> replication rate of a strand depend on the number of catalysts in the
> cell? There is some assumption here about concentrations of strands in
> a cell. It seems to be assumed here that the concentration of strands
> stays fixed when the number of strands goes up. Maybe this is true if
> volume is proportional to the number of strands.

We assume that the concentrations of particles in a protocell is independent of the cellular volume in the main model (we added this information in the revised manuscript; lines 556-557, page 14; lines 588-589 page 15). The minimal model abstracts away the process of chemical reaction and, therefore, dispenses with the concept of concentrations.

> I don't think the model in supplementary section 1.1 adds to the paper
> very much, and given the potential confusion with the main model, it
> might be better to remove this section.

We chose to keep the description of the minimal model in the revised manuscript for the following two reasons. First, this model verifies that the hypothesized minimal conditions are indeed sufficient for symmetry breaking, thereby extracting the essence of our theoretical work. Second, the model is abstract and, therefore, devoid of many specific assumptions only applicable to protocells in the RNA world. Therefore, the conclusions drawn from it are likely to be applicable to a wider context than that of the RNA world. We think these points significantly reinforce the significance and generality of our conclusions.

Moreover, we think we have minimized the chance of confusion in the revised manuscript by re-writing the first paragraph of Supplementary Methods as described above.

> Is there any experimental information that tells us how big V should

> be in a protocell. Having $V =$ hundreds or thousands seems a lot to
> me. I had always imagined $V = 10$ or so.

The experimental study of Bansho et al. (2016 PNAS) cited in the manuscript reports that the average concentration of host RNA molecules in protocells oscillates between $\sim 1e-3nM$ to $\sim 1e+3nM$, and that of parasitic RNA molecules oscillates between $\sim 1e-2nM$ to $\sim 1e+5nM$ (these values were measured after the incubation before transfer to a fresh medium). Their experiments employed water-in-oil emulsion with droplets of 1 micro meter radius on average. Under the assumption that the droplets are spherical, the average number of host molecules per droplet oscillates between $\sim 1e-3$ and $\sim 1e+3$, and that of parasites between $\sim 1e-2$ and $\sim 1e+5$. Therefore, V seems to be more than thousands (with both hosts and parasites included). If V is even greater, i.e. if the experimental setup allows greater concentrations, protocells become evolutionarily unstable and go extinct (Bansho et al. Chem Biol 2012, cited in the manuscript). Moreover, in the experiments of Bansho et al. (2016), the droplets were homogenized at each step of the serial transfer, so that replicators underwent repeated cycles of mixing. This possibility of mixing, which is ignored in our model, would increase the chance of parasites' appearing in a protocell and, therefore, implies that the value of V should possibly be considered effectively higher in the experiments of Bansho et al. That said, we still lack definite information about the right ballpark of V because the maximum allowable number of molecules per protocell is also expected to depend on the mutation rate (Takeuchi, Kaneko, Hogeweg 2016 Proc Roy Soc B, cited in the manuscript), besides biophysical, non-evolutionary factors.

REVIEWERS' COMMENTS:

Reviewer #1 (Remarks to the Author):

Review of resubmission

Manuscript: NCOMMS-17-02243A The origin of a genome through spontaneous symmetry breaking by Takeuchi et al.

General

This revision is much stronger and the manuscript will be interesting for anyone interested in origin of life theory. The most fundamental changes have been to the terminology / language and the assumptions / parameter settings in the model. The addition of the model component illustrated in supplementary figure 3 is very helpful and neatly demonstrates multilevel selection conflict as the cause for symmetry breaking. In fact the new revision in general is an exciting contribution to this field. The authors' interpretation of the "gene" has been articulated. The switch to genome from gene is broadly-speaking more correct, although see comment 1 below. There are a few minor points that would benefit from further qualification. I note the statement that the authors highlight: "symmetry breaking (i.e. the average values of k_{xy} are not invariant under the exchange of P and M)". This clarifies the issue.

Specific comments

1.

In the abstract the term "genome-like molecules" is used. I do prefer this phrase in this particular sentence and understand, given the previous version, why the authors have opted for this. But I don't think using the word genome always captures the essence of what they saying. I apologise for being so pedantic although the authors have specifically asked for my opinion on this. In the context in which the authors are working I don't think there is a term used universally that would probably satisfy everyone. So I would suggest using something like "proto-genomes" (rather in the same way the authors use "protocells") in some instances and where appropriate. For example in the second line of the abstract they ask the question: how did genomes originate? The term genome here implies a genome as we currently understand it, but the genomes they are referring to are fundamentally different. This has special relevance for the transition from genes to genomes. To be safe, they really mean proto-genomes, which could mean a variety of things, so perhaps this term is a safe option. I think in the title it would also be safer to use either "proto-genomes" or "primitive genomes" or "primordial genomes". In other parts of the manuscript it may be fine to use "genes", "genomes", "proto-genomes" etc depending on the context.

I appreciate that to some readers the subtlety may be either lost or frustrating. I could elaborate at a later stage but to philosophers and those investigating levels and units of selection there are some fundamental issues at stake. I would be happy to engage with the authors at another time concerning this point and I have waived my right to anonymity as a reviewer.

2.

In the rebuttal letter the authors state that "...the present study focuses on these two particular features shared by all genomes, implying that we are not concerned with the concept of "molecular biology genome" in its entirety (lines 25-27, page 1)." Yes, this does clarify things. It may also be important to add that their idea of a genome includes the dynamics between component parts – similar to Heng's genome-centric concept (The genome-centric concept: resynthesis of evolutionary theory. *Bioessays*. 2009 May;31:512-25). The model is certainly rooted in the dynamics between strands as well as between levels of selection.

3.

The observation that the molecular structural trade-off (folded vs unfolded) is not directly assumed, but rather emerges, is very satisfying. It provides the theoretical support for the conceptual-theoretical works cited (Takeuchi & Hogeweg, 2007 J Mol Evol and Durand & Michod, 2010, Evolution). This is clear when one looks closely at the model parameters but it may be worth stating it in words as well.

4.

The addition of a one-step replication model works well as a control to test the hypothesis that conflicting multilevel evolution is a key cause of the symmetry breaking. The finding that no trade-off emerges is a key finding. In the legend to suppl fig 3 the statement starting "In the selection algorithm....." is confusing. I think you mean that the second and third particles were chosen from the same protocell as the first particle?

Reviewer: PM Durand

Reviewer #2 (Remarks to the Author):

Thank you for the detailed responses to my previous questions. Considerable work has gone into making the revised version. I think this is a very novel and interesting paper. I recommend acceptance of the revised version.

Reply to Reviewer 1's comments:

We have revised the manuscript based on the reviewer's further comments as described below. In addition, we have made several minor modifications to the manuscript in order to meet the editorial requests. In this submission, we are again providing two versions of an identical manuscript, with and without all changes highlighted (please see Related Manuscript Files).

> Reviewer #1 (Remarks to the Author):

>
> General comments
>
> This revision is much stronger and the manuscript will be interesting
> for anyone interested in origin of life theory. The most fundamental
> changes have been to the terminology / language and the assumptions /
> parameter settings in the model. The addition of the model component
> illustrated in supplementary figure 3 is very helpful and neatly
> demonstrates multilevel selection conflict as the cause for symmetry
> breaking. In fact the new revision in general is an exciting
> contribution to this field. The authors' interpretation of the
> "gene" has been articulated. The switch to genome from gene is
> broadly-speaking more correct, although see comment 1 below. There are
> a few minor points that would benefit from further qualification. I
> note the statement that the authors highlight: "symmetry breaking
> (i.e. the average values of k_{xy} are not invariant under the exchange
> of P and M)". This clarifies the issue.

> Specific comments

>
> 1. In the abstract the term "genome-like molecules" is used. I do
> prefer this phrase in this particular sentence and understand, given
> the previous version, why the authors have opted for this. But I
> don't think using the word genome always captures the essence of what
> they saying. I apologise for being so pedantic although the authors
> have specifically asked for my opinion on this. In the context in
> which the authors are working I don't think there is a term used
> universally that would probably satisfy everyone. So I would suggest
> using something like "proto-genomes" (rather in the same way the
> authors use "protocells") in some instances and where
> appropriate. For example in the second line of the abstract they ask
> the question: how did genomes originate? The term genome here implies
> a genome as we currently understand it, but the genomes they are
> referring to are fundamentally different. This has special relevance
> for the transition from genes to genomes. To be safe, they really mean
> proto-genomes, which could mean a variety of things, so perhaps this
> term is a safe option. I think in the title it would also be safer to
> use either "proto-genomes" or "primitive genomes" or primordial
> genomes". In other parts of the manuscript it may be fine to use
> "genes", "genomes", "proto-genomes" etc depending on the
> context.
>
> I appreciate that to some readers the subtlety may be either lost or
> frustrating. I could elaborate at a later stage but to philosophers
> and those investigating levels and units of selection there are some
> fundamental issues at stake. I would be happy to engage with the
> authors at another time concerning this point and I have waived my
> right to anonymity as a reviewer.

As the reviewer states, the genome as we currently understand it differs from the "genome" we are referring to in the title. In particular, the word "genome" implies more than functional asymmetry and copy-number asymmetry, the two features of a genome with which our study is concerned. To imply this difference, we have inserted the word "primordial" into the title, following the reviewer's suggestion. But, we would like to note here that the above two features of a genome are fundamental as they are universal among all cells. An important implication of our study is that the universality of these features might go beyond this level because they could emerge even in self-replicating molecules, arguably the simplest conceivable system of heredity, if the system is subject to conflicting multilevel evolution.

> 2. In the rebuttal letter the authors state that "...the present
> study focuses on these two particular features shared by all genomes,
> implying that we are not concerned with the concept of "molecular
> biology genome" in its entirety (lines 25-27, page 1)." Yes, this
> does clarify things. It may also be important to add that their idea
> of a genome includes the dynamics between component parts "similar to
> Heng's genome-centric concept (The genome-centric concept:
> resynthesis of evolutionary theory. Bioessays. 2009
> May;31:512-25). The model is certainly rooted in the dynamics between
> strands as well as between levels of selection.

We think that the functional differentiation and copy-number

differentiation clearly entail and, therefore, automatically imply dynamics between component parts. Our study, however, is not directly concerned with such dynamics in general. Thus, for simplicity and focus, we chose not to add the suggested remark to the manuscript.

> 3. The observation that the molecular structural trade-off (folded vs
> unfolded) is not directly assumed, but rather emerges, is very
> satisfying. It provides the theoretical support for the
> conceptual-theoretical works cited (Takeuchi & Hogeweg, 2007 J Mol
> Evol and Durand & Michod, 2010, Evolution). This is clear when one
> looks closely at the model parameters but it may be worth stating it
> is words as well.

Based on the reviewer's suggestion, we have stated that the trade-off inevitably emerges in our model (Line 87, Page 3).

> 4. The addition of a one-step replication model works well as a
> control to test the hypothesis that conflicting multilevel evolution
> is a key cause of the symmetry breaking. The finding that no trade-off
> emerges is a key finding. In the legend to suppl fig 3 the statement
> starting "In the selection algorithm....." is confusing. I think you
> mean that the second and third particles were chosen from the same
> protocell as the first particle?

Yes, that is what we meant. To increase the clarity of the model description, we have expanded the caption of Supplementary Fig. 3.

> Reviewer: PM Durand